



# Technical note: a new analytical protocol for apatite (U-Th)/He and trace element analysis (incorporating a continuous ramped heating measurement system for the He)

Alexis Derycke[1,2]*, Nathan Cogné[2], Dominique Bavay[2], David Vilbert[2], Lionel Dutruch[2], Sebastien Ternois[2], Gilles Ruffet[2], Marc Jolivet[2] and Kerry Gallagher[2]

[1] CRPG, CNRS, Université de Lorraine, UMR7358, F-5400 Nancy, France
[2] Géosciences Rennes/OSUR, Université de Rennes, France

*Correspondence to*: Alexis Derycke (alexis.derycke@hotmail.com)

**Abstract.** This study details the procedures developed between 2021 and 2023 at the Géosciences Rennes lab (GeOHeLiS plateform, Rennes University, France) for obtaining precise (U–Th)/He thermochronological data on apatite. The methodology involves crystal selection, measuring helium ($^4$He) content by heating crystals, and analyzing them using noble gas mass spectrometry. Additionally, determining U, Th, and Sm contents is achieved through crystal dissolution and subsequent solution analysis with triple-quadrupole mass spectrometry (QQQ-MS). Emphasis is placed on a new approach to

quadrupole $^4$He sensitivity calibration based on Durango apatite, ensuring a precision of ~3.9% in $^4$He content determination. The study also highlights the development of a "ramped heating – direct analysis" protocol used to screen apatite diffusion behaviour. In addition, a new quantification protocol for major and trace elements measured in apatite is presented, employing a standard concentration range approach rather than the isotopic spiking technique commonly used for (U-Th)/He studies.

**Short summary:** This note details a development of the (U-Th)/He method for dating apatite crystals in low-temperature thermochronology. Our innovation simplifies analysis protocols for gases (He) and crystals (U, Th, etc.), aiming to expedite data production and enhance study reliability.

## 1    Introduction

After the first attempts in dating rocks by Rutherford in the early 1900s, the development of the (U-Th)/He dating method was revisited during the 1960s (Damon and Green, 1963) around the same time as the emergence of the concept of thermochronology (Dodson, 1973). After a pause in interest, (U-Th)/He analysis was reintroduced as a potential thermochronometrical tool by Zeitler et al. (1987) and has actively evolved since then (e.g. Reiners and Ehlers, 2005; Ault et al., 2019 and references therein). Since the 2000s (U-Th)/He analyses are classically performed on various crystalline





systems (e.g. apatite, zircon, Fe oxides) and become a key method to reconstruct thermal histories of the upper (< 10 km) continental crust. The increase in the amount of data produced and the need to interpret more complex data sets led to an improved understanding of the (U-Th)/He system. This has been particularly manifested by progress in models of diffusion, allowing for the effects of radiation damage (Flowers et al., 2009; Gautheron et al., 2009; Gerin et al., 2017; Willett et al., 2017). These more theoretical aspects have also been accompanied by technological advances allowing the recent

development of new analytical procedures (Idleman et al., 2018; McDannell et al., 2018; Ault et al., 2019; Guo et al., 2021). As part of this recent technical development, this contribution presents (i) the analytical capacity and methodology built between 2021 and 2023 at the GeOHeLiS plateform, Geoscience Rennes, Rennes University, for conventional AHe dating, together with systematic trace element content characterisation of single grains and (ii) the development of a "ramped heating – direct analysis" protocol. The objective behind those developments is to provide tools to assist/improve

interpretations of AHe data by obtaining a chemical composition for each individual analysed grain and the potential to estimate grain specific diffusion kinetics. In detail, the analytical procedure we present exploits an in-house noble gas extraction-purification line coupled to a quadrupole mass spectrometer and internal chemical analysis procedure based on a QQQ-MS spectrometer. All the data reduction performed for routine analysis presented in this contribution were developed using a custom software based on Excel® and VBA and that can be distributed upon request.

## 2    Noble gas line

The sample degassing, gas purification and helium quantification are done on the in-house noble gas line connected to a quadrupole mass spectrometer (Fig. 1A), from now on referred to as the Q-He line. The Q-He line is composed of several sections (½ inch Swagelok®) isolated by pneumatic Swagelok® valves coupled to electro-valves (ASCO/JOUCOMATIC™). The electro-valves and some other components (heating cable, furnace and quadrupole) are

controlled by an electronic switch card (USB SSR-24, Measurement Computing™) USB piloted through LabVIEW®. The Q-He line can be sub-divided into three sections (Fig. 1A):

   (i)      the cell and heating system
   (ii)     the "cleaning" section where the vacuum is maintained by a turbo-molecular pump (Hi Pace 80, Pfeiffer Vacuum™) coupled to a primary pump (MVP 070-3, Pfeiffer Vacuum™)

(iii)    the "spectrometer" section where the vacuum is ensured by an ion pump (Vaclon Plus 40, Agilent Technology™)





**Figure 1: Schematics of (A) the Q-He line and (B) the heating system connected to a laser generator through an optical fibre.**

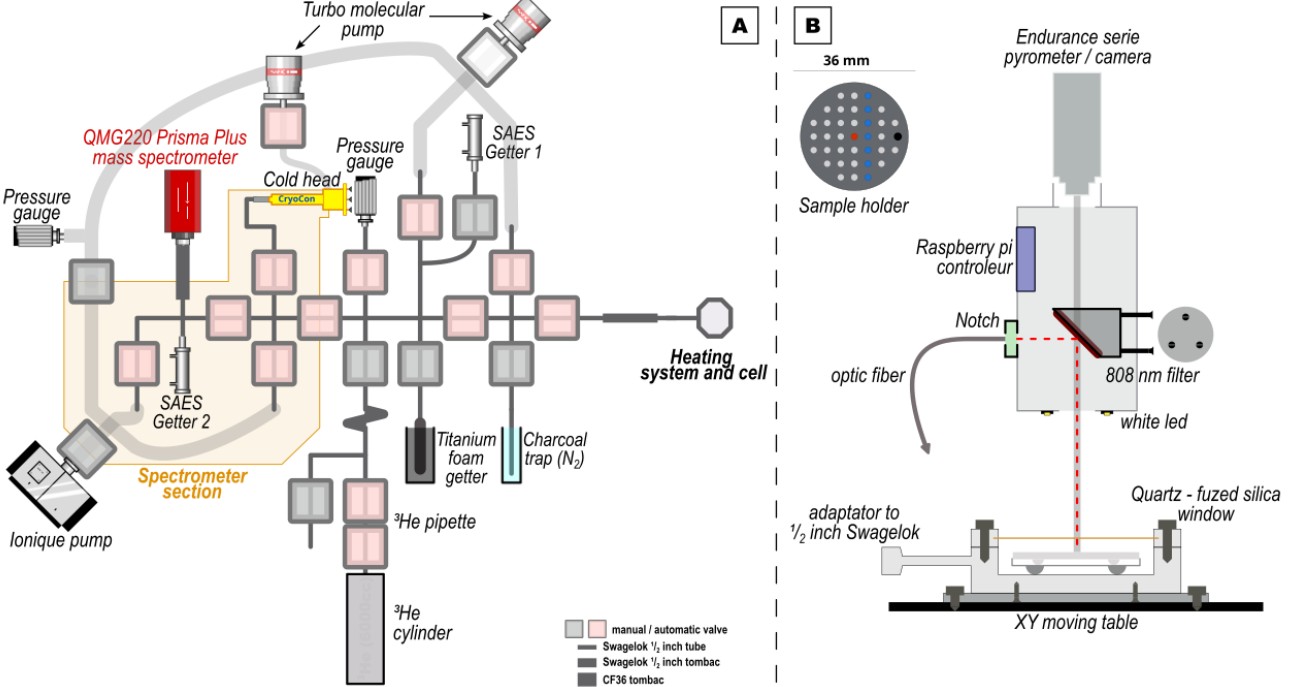

## 2.1 Heating system:

The heating system (Fig. 1B) is composed of a cell (quartz/fused Silica viewports CF36 flanged, 550-1000nm from Kurt Lesker™) fixed to a XY motorised support (OptiScan® III – 1µm unit, Prior Scientific™) and computer controlled thanks to LabVIEW® assets provided by manufacturers. This support allows us to move sample holders under a static heating beam. The heating is ensured by an infra-red diode laser (nLight Pearl TKS – 808±3nm – 40W, SAKAR Technology™) coupled to an optic fibre, an electrical lens (ELD 4, Optotune AG®) and an optical filter (Notch 808nm, Edmund Optic™). The capsule temperature is monitored with a two-wavelength pyrometer (Endurance® Series – E2RL-V0-0-0, Fluke Process Instrument™) including a camera that allows visual confirmation of the sample holder positions. In detail, the laser TKS provides the power control (from 0 to 40W, ± 0.1W) and the laser duration/pulse timing (from 10±1µs to continuous) used to regulate the transmitted power. The Endurance® pyrometer has a temperature range between 250°-1200°C on all types of metallic materials in a restricted area over a 2mm diameter surface (with a focal distance between 19 and 30 cm).

Practically the 808nm optic filter refracts the laser and provides a control for the beam position (Fig 1B), whereas an electrical lens is used to define the laser focus (~1mm diameter). The pyrometer is installed on the other side of the 808nm optic filter (vertically) and targets the sample though this filter. Pyrometer temperature measurements can nonetheless be perturbed by the absorption of visible light by the cell window and the 808 nm optic filter. This is autonomously handled by the Endurance® system and can be reduced by increasing the time of light integration for a temperature measurement. The significance of those temperatures will be discussed in sect. 4.3.1. This global heating system was made in-house and



completely interfaced/controlled by LabVIEW® (some assets provided by manufacturer, others in-house). The automation, combined with the laser and pyrometer capabilities and including a calibrated PID (proportional–integral–derivative system), allows us to implement various heating schedules (e.g., constant temperature, ramped or multi-stage heating) with precise

temperature response and control.

### 2.2     Gas cleaning material

Several traps are set up on the Q-He line for gas cleaning: x1 active charcoal on inox tube, x1 titanium foam on inox tube, x2 SEAS® Getter (GP50-W2F with St101 material) and x1 active charcoal on a cold head (CH-202FF, Sumitomo SHI® Cryogenic of America™) equipped with Silicon Diode temperature sensors (CY7 series®). The cold head has a capability to

reach ~11°K and is connected to a Model32 controller (Cryogenic Control Systems™) interfaced to computers by LabVIEW® (manufacturer asset). The Getter 1 (Fig 1A), in the "cleaning" part, can be operated at room temperature or heated. The Getter 2 (Fig 1A), directly connected to the spectrometer, is always operated at room temperature to ensure a low $H_2$ partial pressure. All traps, excepted Getter 2, can be isolated/connected and activated/deactivated depending on the desired cleaning level.

### 2.3     $^4$He quantification approach

In Q-He line, the gas analysis is performed on a quadrupole spectrometer PrismaPlus® QMG220 (Pfeiffer Vacuum™) with operational pressures between $10^{-14}$ and $10^{-5}$ mbar. The $^4$He quantification is based on previously published methods using a $^3$He spike (House et al., 2002; Farley, 2002) and subsequently modified by Gautheron et al. (2021). In the firsts publications, a spike was used to ensure a constant pressure in the mass spectrometer during the gas analysis and determine the $^4$He

amount by peak comparison given that the spike amount is precisely known. To quantify the $^4$He amount, Gautheron et al. (2021) used a continuous calibration of quadrupole sensitivity (based on standard ages) combined with a $^3$He spike correction. In our implementation, we built on the Gautheron et al. (2021) method, simplifying it a little and modifying it to allow preforming continuous gas analyses during ramped heating experiments developed by Idleman et al. (2018).

For the Q-He line, the spike is made with a 5741± 24 cc (1σ) inox cylinder (in-house) and a 5.8±0.1 cc (1σ) pipette made by

two Swagelok® valves (connected by a "male-male" Swagelok® and stainless-steel tube placed inside to reduce the volume). The cylinder is (re)filled with pure $^3$He gas from an external 1L tank (isotopic purity >99.9%, GVL Cryoengineering®). The internal cylinder pressure is adjusted to the desired analytical pressure in the Q-He line (cleaning + cell volume), ca. 1.0 $10^{-6}$ mbar (~5x$10^{-11}$ mol). The amount of $^3$He injected by the pipette will decrease following equation 1, (Gautheron et al., 2021), and this can then used to predict the decrease in $^3$He signal (measured in amperes).

$$^3He_{predicted}(pipette\ number) = {}^3He_{initial} \times \left(\frac{Vol_{cylinder}}{Vol_{cylinder} + Vol_{pipette}}\right)^{pipette\ number} \qquad (1)$$





The predicted $^3$He signal is then compared to the measured $^3$He signal to determine an "ionisation efficiency" parameter ($I_e$ in Eq. 2). $I_e$ is adjusted for each analytical session (values routinely range between~0.7 and ~1.3.) to incorporate the evolution/effect of external factors (such as room temperature instability, power instability or failure, volume changes) and allows the calculation and use of a constant quadrupole $^4$He sensitivity through time ($S$ in Eq. 2, ccSTP/amp or mol/amp). Note that $I_e$ could be compare to the $D$ parameter presented in Gautheron et al. (2021) but influence jointly the spike amount and the sensitivity. This adaptation from the Gautheron et al. (2021) formulation was motivated by analytical feedback and results in a simplest use and explanation of this parameter. Thus, the $^4$He/$^3$He ratio is calculated by dividing the $^4$He signal (in amperes) by the $^3$He signal (in amperes) corrected of the HD+ isobaric contribution. Then the $^4$He/$^3$He ratio of the base line, automatically selected between the last reheat or the last blank in the series (see Gautheron et al. (2021) for discussion), is subtracted from the $^4$He/$^3$He ratio of the sample. Finally, the $^4$He quantification is calculated using the following equation:

$$^4He_{sample} = \left(\frac{^4He_{signal\ sample}}{^3He_{signal\ sample}} - \frac{^4He_{base\ line}}{^3He_{base\ line}}\right) \times {}^3He_{predicted}(pipette\ number) \times S \times I_e \qquad (2)$$

### 2.4 Quadrupole sensitivity determination

As our formulation of the $^4$He calculation differs from Gautheron et al. (2021), the determination of quadrupole sensitivity was adapted. It was made using the approach proposed by Solé and Pi (2005) initially developed for magnetic sector MS but which we have adapted to our analytical setup (i.e. quadrupole MS). After any quadrupole tune, the sensitivity – $S$ is (re)calibrated by analysing ~20 fragments of Durango standard, covering a large range of sizes, following the protocol presented here (see sect. 4.2). Then the U-Th-Sm contents and standard ages are used to determine the amount of $^4$He theoretically present in each fragment. The theorical $^4$He content (ccSTP or mol) can therefore be plotted versus the corresponding $^4$He mass spectrometer signal (amperes) and used to calculate the mass spectrometer sensitivity – $S$ (ccSTP or mol /amp) and precision. During this calibration, Durango standard is actually used as a container with a known volume of $^4$He, replacing the classically use of volume calibrated $^4$He pipette (see Solé and Pi, 2005).

Later, during routine analyses Durango standard are continuously analysed to constantly monitor the $I_e$ and more broadly to make sure the Q-He line is running as expected (Gautheron et al., 2021 see sect. 4.2 for discussion). This approach is completed with a second standard analyses thanks to MK-1. In this combination, Durango standard correspond to a primary standard (use to tune the $I_e$ parameter based on its known age) and MK-1 to a secondary one, analysed as an unknow sample, and therefore used to check reliability of $S$ and $I_e$ parameters.

### 3 (U-Th)/He protocol

### 3.1 Samples preparations, apatite pickings and packings

Apatite rich powder is obtained with conventional separation procedures, including mechanical crushing, sieving and mineral density/magnetic separation. Apatite hand picking is performed under a binocular microscope (Olympus™ SZX16,



x6 to x111) equipped with reflected/transmitted light, polarised light and a camera (an Olympus™ DP20-E or a ToupCam CMOS 18Mpx from ToupTek™). The light capacities (transmission/reflexion and polarised/analysed) allow the observation of alteration/inclusion in crystals. Cameras are connected to a computer and grain/crystal size can be measured using (i) in-house LabVIEW™ software (Olympus™ DP20-E) that includes automatic grain detection and size measurement, or (ii) ToupView™ (ToupCam) software. For routine analyses, apatite crystals are selected using conventional criteria (size, morphology, inclusion free). Two age standard materials are also used, Durango (McDowell et al., 2005) and MK-1 (Wu et al., 2021). For Durango, centimetric apatite crystals are gently crushed in an agate manual mortar until the obtention of a µm-mm scale powder. Durango has an age of 31.02±1.01 Ma (McDowell et al., 2005) while  MK-1 is used as a slightly younger standard (18±0.7 Ma, Wu et al., 2021). For both standards, various sizes of fragments (50-500 µm) are selected to constantly assess the quadrupole mass spectrometer sensitivity and blank levels. Selected crystals or fragments are then packed in platinum tubes "capsules" (99.95% purity, 1.0x1.0 mm from AMTS™) that have already been folded and placed in a parafilm bracket.

### 3.2    $^4$He analytical protocol

Capsules containing the grains are placed in a custom inox sample holder (which has 35 positions that are optimised to decrease air traps) and then loaded in the Q-He line cell. Following loading, the Q-He line is heated to 50°C and pumped until it returns to the routine pressure ($< \sim 1\ 10^{-8}$ mbar). Samples, standard and blank are then analysed using either conventional or "ramped heating – direct analysis" protocols. Our protocols are built on previously published ones (Gautheron et al., 2021; Idleman et al., 2018) that have been adapted to our experimental installation.

### 3.2.1    Conventional $^4$He analyse

For a conventional analysis, the cell is firstly isolated from the line during the sample heating (5 minutes at 1050°C for Apatite), then the extracted gas is mixed with one pipette of $^3$He and cleaned following the desired protocol. For apatite the cleaning protocol corresponds to 10 min in contact with the Getter 1 and the cold head operating at 30°K to maximise gas trapping except for $^3$He and $^4$He (Fig. 2). Given the absence of hydrocarbon pollution in apatite crystals, Getter 1 is operated at room temperature to maximise the H$_2$ trapping.





**Figure 2: Quadrupole pressure (in amp.) for different masses depending on the cold head temperature (CH-202FF, Sumitomo SHI® Cryogenic of America™). 30°K maximizes the desorption of 3He and 4He while maintaining the trapping of other masses and corresponds to the routine operating temperature.**


The cleaned gas is then introduced in the mass spectrometer line segment and analysed after 30 seconds of gas equilibrium. The analysis corresponds to 20 cycles of 0.5s per mass. For apatite, we routinely measure not only $^{3}$He (m3) and $^{4}$He (m4) but also $H_2$ (m2), $H_2O$ (m18), $N_2$ (m28), $^{40}$Ar (m40) and $CO_2$ (m44) to quantify the gas pollution. Furthermore, mass 5 is also measured to monitor the spectrometer vacuum quality. After each analysis, the whole Q-He line is pumped for 2 minutes

before dealing with the next sample, and the cold head is regularly (every 20-25 sample) heated to 140°K and pumped for 30 minutes to clean it.





Each grain is systematically reheated until it is completely degassed, which for apatite should occur during the second heating. The ratios of $^4$He/$^3$He obtained are then uses to calculate the $^4$He amount in the grain using Eq. 2 by comparing it to the last blank (see Gautheron et al. (2021) for discussion).

### 3.2.2    $^4$He ramped heating – direct analyse protocol


For ramped heating – direct analysis, an analysed volume (cell + traps) is firstly isolated from pumping. For apatite, the same traps are used as for conventional analyses: Getter 1 (operated at room temperature) and the cold head (operating at 30°K). Then one pipette of $^3$He is expanded into the analysed volume (cell + traps) and, after 1 min to ensure gas equilibration, the analysed volume is opened to the mass spectrometer. The first 30 seconds after the opening corresponds to a second equilibrium time.


The mass spectrometer measurement, presented in Fig. 3 3, starts ~5 seconds before the volume opening (to monitor the spike introduction, Fig. 3 zone 0) and can continue up to three minutes after the end of the sample heating (to monitor any irregularity gas consumption, Fig. 3 zone 3). The measurement corresponds to a continuous cycle over multiple masses, with a time integration of 0.5s per mass. As for conventional analysis, we measure vacuum quality (m5), $H_2$ (m2), $H_2O$ (m18), $N_2$ (m28), $^{40}$Ar (m40) and $CO_2$ (m44) to quantify the gas pollution and traps efficiency.




**Figure 3: Gas evolution during ramped heating routine analysis (case of a Durango fragment, D22P3-F). The three analysis steps, discussed in sect. 3.2.3, are presented. Part A and B highlight two gas pollution gas blows events observed on most of the analyses and are discussed in sect. 4.3.2.**

The first 30s after spike introduction corresponds to gas equilibrium in the spectrometer (Fig. 3, zone 0) and are characterised by a rapid consumption of major gas ($^3$He, $H_2$ and $^4$He). After this equilibrium, the spike is analysed during 3 minutes before the sample heating starts (Fig. 3, zone 1), these analyses will later be used to determine the blank evolution (see sect. 4.3 for discussion). For apatite, sample are heated from 250-1050°C over 20 min, i.e ~40°C/min) (Fig. 3, zone 2). This heating is automatically handled and includes sample re-centre and PID controlled temperature. Thus, the real time can vary from 20 to 25 min to reach the 1050°C, depending on the capsule reaction to the laser beam. After the end of the



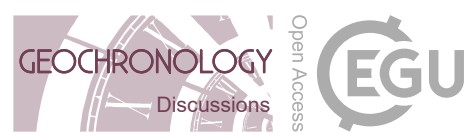

heating, the mass spectrometer measurement continues over 3 minutes to analyse the total gas (Fig. 3, zone 3). The data from this period is subsequently used to quantify the $^4$He content (see sect. 4.2 for discussion).

Between each sample, the Q-He line is completely pumped for ten minutes, a sufficient duration to return to the routine pressure (~1 x10$^{-8}$ mbar). The cold head is heated and cleaned following the same protocol and frequency as for
conventional analysis. Two pollution events, noted A and B in the Fig. 2, occurred during the degassing stage and will be addressed in sect. 4.3.2.

### 3.2.3    Quadrupole mass spectrometer signal process for the ramped heating – direct analyse protocol

To reconstruct the $^4$He degassing spectra and quantitatively estimate its amount, the mass signal is processed following two methods.
The degassing spectrum is reconstructed from the $^4$He/$^3$He ratio rather than using the $^4$He signal. This choice was made to take account of the gas consumption and other potential gas-related perturbations. The ratio measured before the start of heating (Fig. 3 – zone 1) is used as a "blank value", linearly interpolated over time considering a linear evolution (Fig. 4A), and subtracted from all the spectra (Fig. 4B) to calculate the cumulative helium loss (commonly name $f$, Fig. 4C). This correction was made to account for gas consumption over time. and will be further discussed in section 4.3.3. The
cumulative helium loss is then linked to the sample temperature as a function of absolute time. This correlation can be done without a lag correction due to the fact that laser heating should not induce a lag in sample heating (Idleman et al., 2018), and the gas transport between the cell and the spectrometer is fast (<1 second, McDannell et al. 2018). Finally, as both temperature and $^4$He/$^3$He ratio are raw experimental signals, the spectrum is often noisy. A filter is therefore applied on it to decrease the noise impact.





**Figure 4: Example of $^3$He/$^4$He blank determination, correction, and total degassed fraction calculation (Durango fragment, D22P3-F)**



The $^4$He content is calculated using $^4$He/$^3$He ratio measured during the last 3 minutes of analysis (after the end of heating (Fig. 4C). The interpolated blank value is then subtracted from this measure (Fig. 4A) and the result is used to calculate the

total amount of $^4$He which, in turn, is used to calculate AHe age (Eq. 2 see sect. 2.2).

### 3.3    Crystal digestion and elemental characterisation

After the degassing, capsules are retrieved and put in individual 10mL vials (INFO) for chemical digestion and elemental quantification. This quantification is done by signal comparison to a standard solution ranges with adapted dilution to obtain concentrations in the range from 0.1 ng.l$^{-1}$ (ppt) up to 10 000 µg.l$^{-1}$ (ppb).

For apatite, the digestion protocol is adapted from Farley (2000) and Gautheron et al. (2021) and is performed by 100µl acid attack (HNO$_3$ at 5N or 27%, – distilled from HNO$_3$ 65N – Normapur® VWR®) on a hot plate set to 65°C during 3 hours. After 30 minutes of cooling, solutions are then completed for QQQ-ICP-MS analysis with a HNO$_3$ – 0.5N or 2% solution (distilled from HNO$_3$ 65N – Normapur® VWR®) to 2 to 10 ml depending on the required concentration and analytical protocol. The micropipette used for the acid attack and dilution come from Eppendorf Research Plus® respectively 100-

1000µl and 0.5-5 ml. After the analysis the platinum capsules are retrieved and returned to the supplier for recycling.

To calculate the dilution factor of each sample during the digestion protocol we use the micropipette volumes and uncertainties. For each chemical session, the micropipettes volumes are check by randomly weighting 10 samples after dilution. The reliability of this approach is discussed in sect. 4.1.

The elemental characterisation in solution is made on an Agilent 8900 QQQ-ICP-MS. In addition to U, Th and Sm used for

helium age calculation, all REE elements are routinely analysed to provide complementary information for interpretation on each grain (inclusion, sample homogeneity, source for detrital samples). For apatite, Ca is also systematically analysed to determine apatite weight, considering Ca is stochiometric and a fluorapatite composition (Eq. 3, Gautheron et al., 2021).

$$Apatite_{weight} = \frac{Ca_{weight}}{0.3974} \tag{3}$$

To calculate the element content for each individual grain, measured elements concentrations are then multiply by the

dilution factor.

## 4    Results and discussions

### 4.1    REE and Ca chemistry and blank

Use of standard concentration ranges to quantify the amount of element in a solution is classic and well documented approach in geochemistry. However, practical issues made this generally unsuitable to quantify U, Th and Sm in

thermochronology. The main concern is the size of the apatite grains (generally under ~200µm) and the often-low concentration of these elements (0-500 ppm U, 0-1000 ppm Th). These two factors and the requirement to dissolve and dilute the grains in a significant amount of solution (for IC-MS analysis) meant that the concentrations were usually below





the sensitivity of the ICP-MS. One solution to this problem is to use isotope dilution (Evans et al., 2005; Guenthner et al., 2016) and this has been used for AHe and other thermochronometric methods. However, a current practical concern

(increasing difficulty to obtain isotopic standards, particularly $^{230}$Th) and the improvement of the ICP-MS resolution motivated us to reconsider the use of standard range comparisons. In assessing this, we ran the following evaluations to validate it as viable protocol.

### 4.1.1 Blank purity

The Fig. 5A presents the signal obtained on solutions analysed in different chemistry sessions. The acid-blanks (black line)

are characterised by an utra-pure signal, with less than 0.1 ng/l (or ppt) signal at the limit of resolution. Nevertheless, capsule-blanks (grey line) show a small but systematic contamination in REE. For most of the elements this contamination is significantly lower than signal observed in samples (1-2 orders of magnitude lower) and is therefore considered as negligible. However, for the critical elements such as the U and Th, this contamination may reach ~10% of the total signal, with a signal at ~10 ng.l$^{-1}$ and a contamination at ~1 ng.l$^{-1}$ (Fig. 5). The highest contamination appears with Th with a blank

signal at ~0.6 ng/l (ppt) and a capsule-blank signal at ~2.12 ng/l (ppt) signal. This can be explained by the documented pollution produced by the platinum capsule on Th attributable to the affinity that exists between Th and Pt.

**Figure 5: Evaluation of the blank purity for the peak comparison protocol. A: concentration in solutions over 5 analysis sessions. "Sample blank" corresponds to capsules that should have been loaded with a crystal but contain no Ca in analyses and so were interpreted as empty. B: distribution of U, Th, Sm and Ca elements in all "blank" (platinum blank, chemical blank and sample**

**blank) used to define the "chemical blank" subtracted to obtain the estimated element concentrations in a crystal (sect. 4.1)**

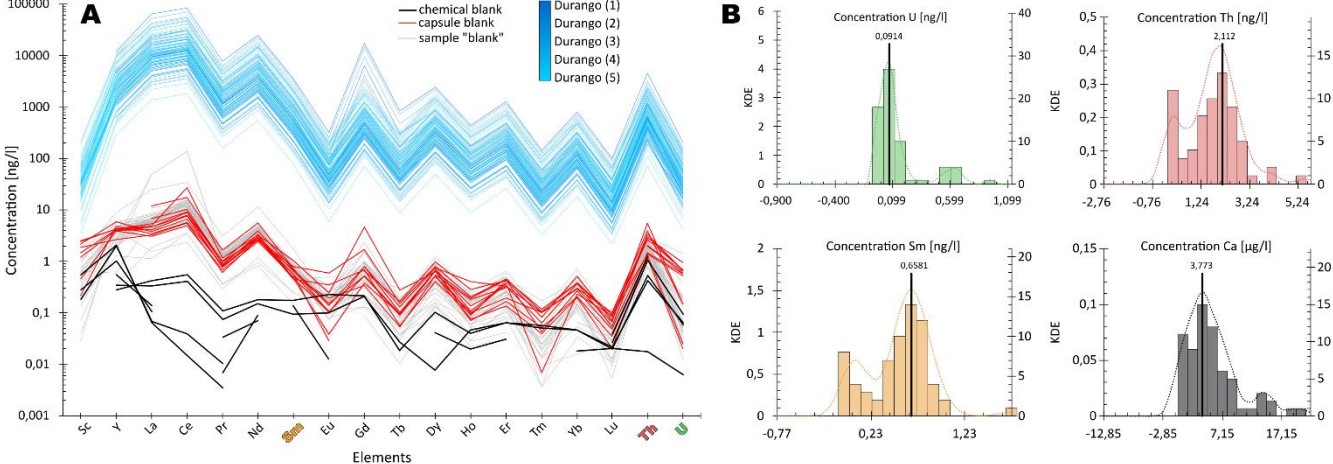

Given this contamination problem and knowing the variability of element concentrations observed in analysed crystals (0.1 to >100 ppm), we decided to determine a capsule-blank mean value for all elements (Fig. 5B), and systematically subtract it when analysing crystals. As this contamination is not perfectly reproducible, it affects the measure and precision of

elemental quantification for crystals with low concentrations. To identify such crystals, we define an uncertainty of 25% for the capsule-blank mean values. Crystals whose signal falls in this uncertainty range are therefore rejected. Fig 6 presents the




minimum concentration needed in a crystal (vs apatite weight and grain size, indicated as equivalent spherical radius, $R_S$) to be measurable by our method for the U, Th and Sm element. We argue that those values are reasonable for most cases, subject to the condition of not selecting apatite crystals bigger than 35-40 µm. Moreover, this size is consistent with crystal

selection for AHe method (i.e. $F_T$ criteria, Ketcham et al., 2011).) .

**Figure 6: minium concentration required in a crystal to be used in the spike comparison chemical protocol versus apatite weight or equivalent spherical radius (Rs). Calculations were made using the blank value of Fig. 5.**

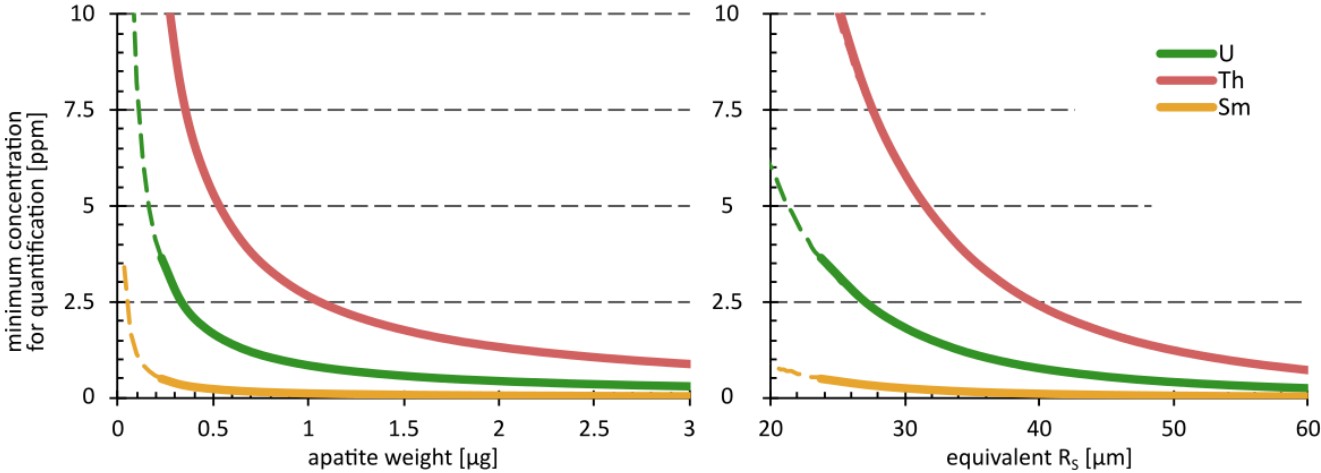

### 4.1.2    Dilution factor consistency and concentration calculation

In contrast to isotope dilution, the precision of dilution factor is the key point to be able to properly quantify the elemental abundance in analysed crystals, and then calculate their concentrations. To quantify its reliability two different approaches were used. Firstly, we compare our observed dilution error to a predicted one. These are respectively determined by weighing each dilution and using the manufacturer specification (100±0.6µL and 5000±7.5µL). Figure 7A presents results

from 5 chemical sessions (yellow) and the value of the micropipette use for the dilution (grey). A systematic shift is observed between the theoretical and measured value, but the relative error is constant. We interpreted this systematic shift as a product of micropipette de-calibration over time (referenced by the manufacturer), and we suggest that is possible to use micropipette precision (0.162%) as the dilution error as long as 5-10 weight measurements are made during a chemical analysis session to account for the de-calibration shift.





**Figure 7: Evaluation of the dilution precision and concentration calculation. A: mean and uncertainty of 5 sessions of dilution (yellow) and theoretical value and error of the micropipette used (grey). B: REE concentration (normalised to Chondrite) of fragments of a Durango crystal (which seems to be internally homogeneous).**

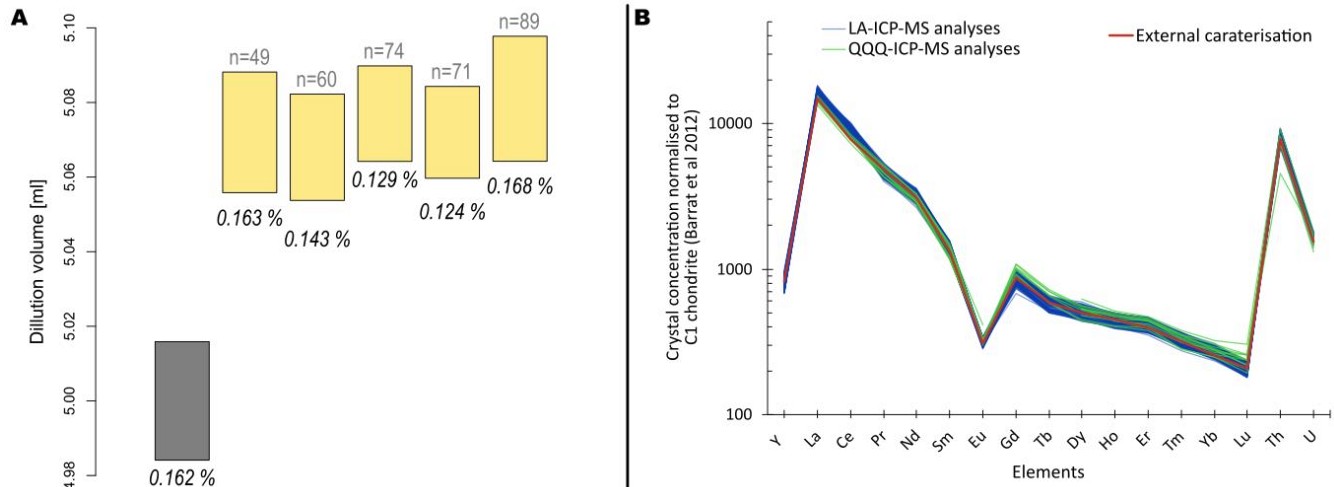

The second approach consists of analysing a well-known internal standard. Here we used a Durango apatite crystal used for
LA-ICP-MS fission track analysis and which seems very homogeneous in terms of U and Th. Fragments of this crystal have been dissolved and analysed in an external laboratory (Service d'Analyse des Roches et des Minéraux - SARM, Nancy, France) giving an independent reference value. Fifteen small fragments of this crystal have been analysed using the protocol described here and the recalculated crystal concentrations are presented in the Fig. 7B together with the SARM analysis (in red) and the LA-ICP-MS data (blue). Those results show a good reproduction of the crystal concentrations for multiple
crystal sizes, confirming the reliability of the calculated dilution factor.

### 4.1.3 Chemical standard

For Durango, the chemistry is usually evaluated with the Th/U (and Sm/Th) ratios (i.e Evans et al., 2005; Guenthner et al., 2016; Gautheron et al., 2021). Figure 8 present those ratios from fragments analysed between 2022 and 2023. The ratios for Th/U and Sm/Th range from 14.9 to 23.0 and 0.62 to 1.17 respectively and are in agreement with published values
(McDowell et al., 2005; Reiners and Nicolescu, 2006; Gautheron et al., 2021). However, the ratios show a negative linear correlation not reported in the previously published Durango datasets (e.g. Gautheron et al., 2021). If this observation was produced by a chemical bias on the Th measurement, we would not expect a linear correlation but a systematic ratio shift. From an internal lab experience, the same relation has been observed on different volcanic apatites. As Durango is a volcanic apatite, we suggest that this relation is a product of the crystal heterogeneity formed during crystallisation rather than
chemical issue. Based on this assumption, crystal not consistent with this correlation have anomalies in their element content characterisation, reflected in anomalous ages (Fig. 8). The origin of those anomalies is unclear, and we propose that is could




came from U, Th or Sm loss during the heating (old ages) or enrichment as a consequence anomalous platinum contamination (young ages).

**Figure 8: Th/U and Sm/Th ratios of Durango colour coded by age for fragments analysed in 2023 for this study, and the same plot made using the dataset of Gautheron et al. (2021)for comparison.**

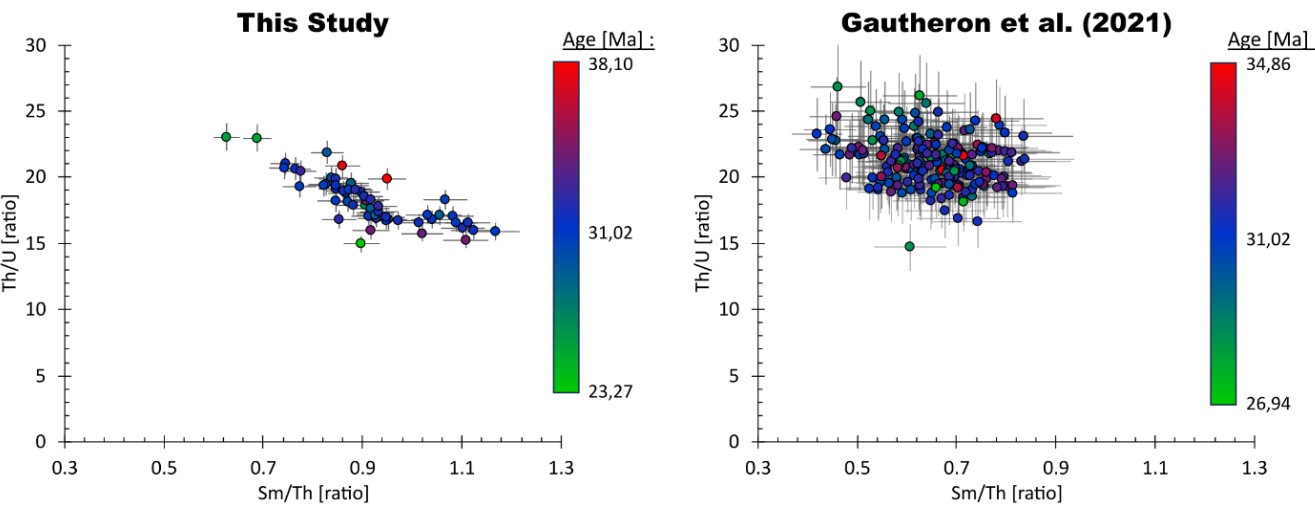

## 4.2    Quadrupole calibration and standard ages

### 4.2.1    Quadrupole calibration

34 Durango fragments of different sizes were analysed to determine both the $^4$He signal (amperes) and U, Th and Sm content (ng) as explained in the previous sect. (see sect. 3). To determine the $^4$He signal we modify Eq. 2 by removing the sensitivity (S) and setting the $I_e$ parameter to 1. The U, Th and Sm contents were then used together with the Durango AHe ages (31.02±1.01 Ma, McDowell et al., 2005) to calculate a theoretical $^4$He content (in ccSTP or mol). Results as theoretical versus measured $^4$He content are shown in Fig. 9A. The main uncertainty of the theoretical $^4$He content, greater than the

measured one, comes from the error on Durango ages (1.01 Ma) used during this calculation. Linear regression gives a p-value of 0.98 (MSWD=0.56, using IsoplotR from Vermeesch, 2018), indicating a good correlation and so a good confidence in the slope and intercept. We therefore propose that the regression slope can be interpreted as the Q-He line sensitivity (*S*). The x value for y=0 (indicating the absence of $^4$He) yields the spectrometer's "electronic blank" signal for $^4$He. Results give a sensitivity of value of 6 142±167 ccSTP/amp and a $^4$He blank signal of ~8x10$^{-15}$ amperes in accordance with the value

observed for routine blank (between 6 and 8x10$^{-15}$ amperes).



**Figure 9: Estimation and use of the *S* and *I_e* parameters for quantifying [4]He content. A: determination of the quadrupole sensitivity *S* using the [4]He signal and the calculated [4]He content. B: impact of different external events on the [3]He signal between September 2022 and July 2023 (pipette number corresponds to successive analyses).**

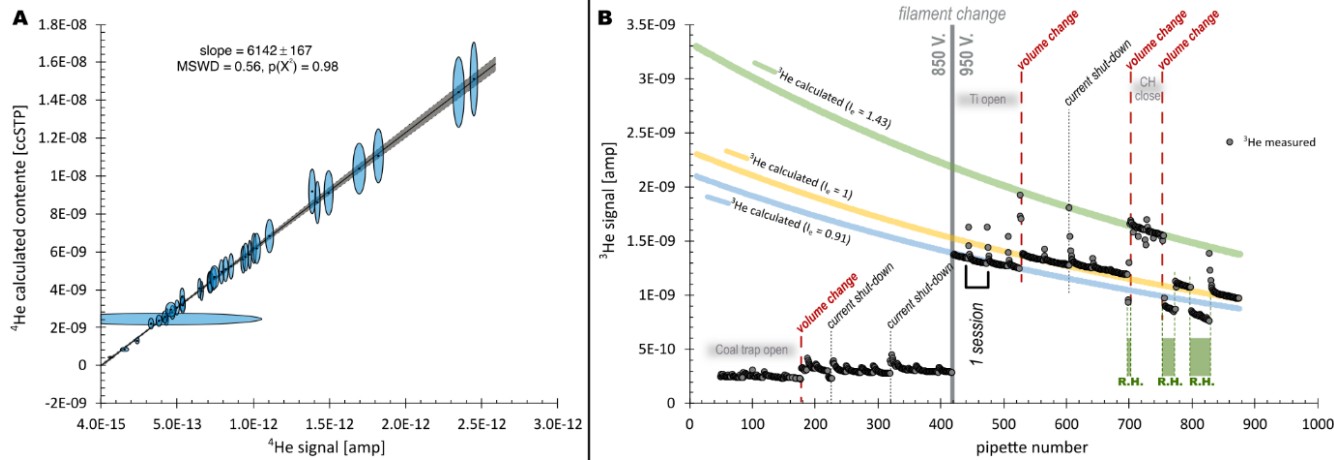

However, as mentioned earlier (sect. 2) the spectrometer sensitivity is function of several parameters (e.g., filament voltage, gas pressure…) that may be modified in a regular or irregular manner (e.g. spectrometer tuning, line modification/maintenance, room temperature, power failure). To illustrate the effect of such changes, Fig. 9B presented the [3]He spike signal variation over pipette numbers (for about a year) together with different events that occurred during this period The $I_e$ parameter (sect. 3) was designed to handle such variations and to illustrate this it is possible to calculate a theoretical [3]He content (Eq. 1), multiply it by $I_e$ and then compare this to the measured [3]He signal. So, Fig. 9B includes the [3]He theoretical signal for three different values of $I_e$ and shows a good prediction for different analytical conditions. We thus argue that using this approach it is possible to use a unique sensitivity value (*S*) as calculated earlier, in case of volume or protocol modification events, as long as the value of $I_e$ is tuned properly.

Finally, although we have used a 34 of Durango shards, we suggest that to calibrate the Q-He line sensitivity appropriately, a dozen of various sized crystals should be enough.

### 4.2.2 Standard ages

Between February 2023 and July 2023, during routine analyses 45 Durango fragments and 10 MK1 fragments were analysed, as primary and secondary standards respectively, during a total of ~500 analyses (pipette 300 to 900, Fig. 10A). The Durango fragments yield a central age of 31.11±0.23 Ma in accordance with the published value of 31.02±1.01 Ma (McDowell et al., 2005). This central age is characterised by a dispersion of 3.9±1.3% and three populations whose the main one include >90% of the data at 31.22±0.15 Ma (Fig. 10B, Galbraith, 1988 radial plot, on IsoPlotR Vermeesch, 2018). We argue that this homogeneity over a long analytical period is evidence of the $I_e$ calibration reliability. The remaining 10% of data correspond to anomalous Durango ages (both older and younger) that seem to correlate to anomalous Th/U-Sm/Th ratios crystals (Fig. 8). We therefore attribute these anomalous ages to U, Th, Sm perturbations rather than an [4]He



measurement issue. For a discussion about the origin of this perturbation, see the sect. 4.1.3. The MK1 crystals yielded a weighted mean age of 17.7±0.6 Ma in agreement with the published value of 18.0±0.5 Ma (Wu et al., 2021). As the MK1 crystals were treated as unknown samples (secondary standard), the age reproduction supports our calibration methodology.

**Figure 10: A: Standard (U-Th)/He ages as a function of pipette number, grey area corresponds to analyse done before a significant Q-He line change. B: Radial plot of the Standard (U-Th)/He ages.**

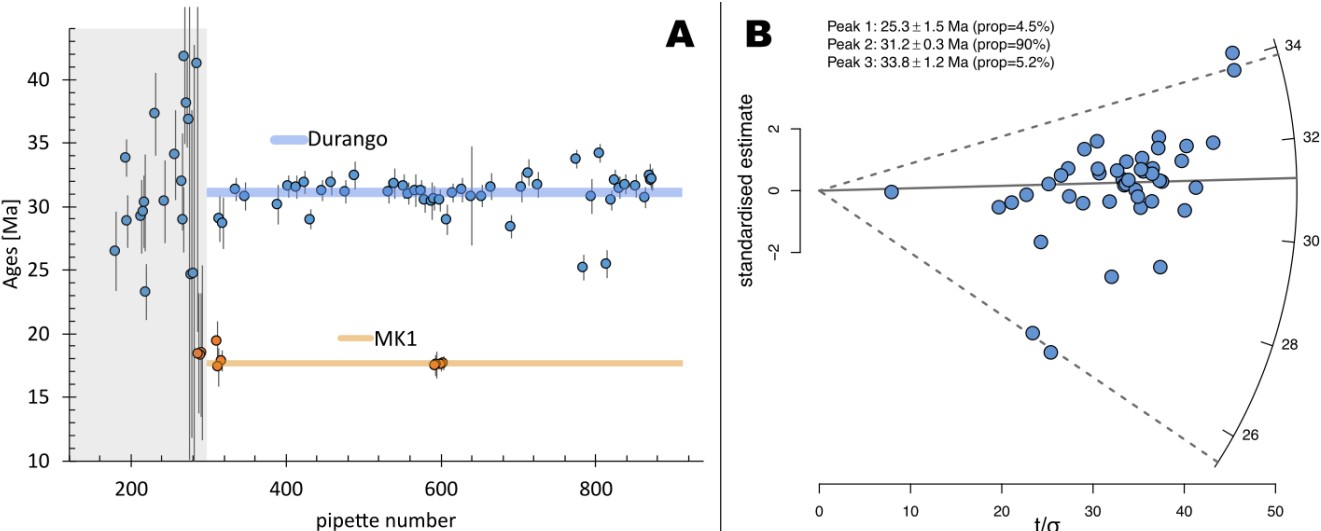

### 4.3 Feedback on the ramped heating development

**4.3.1 Temperature control**

Temperature is a key parameter for diffusion experiments and so needs to be assessed during ramped heating. To test the reliability of our in-house setup (Fig. 1B), we performed a reading/heating comparison of a capsule in the cell at the ambient atmospheric conditions by i) heating a capsule set on a thermocouple (K type - class 2, Fig. 11A) and ii) using the pyrometer Endurance® to control the laser power. As examples, Fig. 11B present the outcome of two experiments for a classical 370 ramped heating and a more complex heating schedule. The results show a good consistency between the temperature estimates, especially above ~350°C. We interpret the significant difference below ~350°C to reflect limitations in the Endurance® pyrometer (min. temperature = 250°C for optimum conditions) and therefore can only use temperature estimates only above this minimum limit. A small difference (~6°C) is observed at temperatures above ~800°C and is interpreted as an effect of the thermocouple energy accumulation.




**Figure 11: Evaluation of the reliability of the Endurance® pyrometer. A: test set-up initiated at atmosphere temperature. B: temperature inferred from the pyrometer (red line) and the thermocouple K (black line) during different heating procedures.**

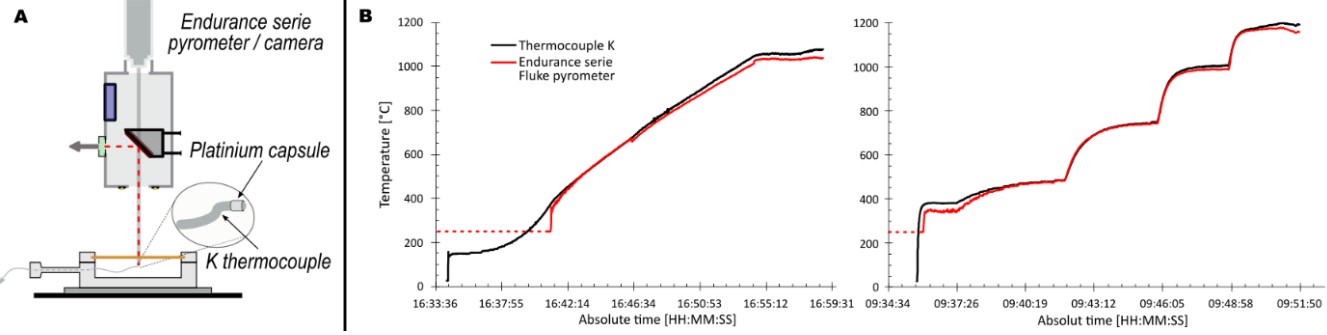

### 4.3.2 Gas pollution

During ramped heating analysis, some systematic gas pollution is observed and is displayed as the grey area "A" and "B" in the Fig. 3. The "A" pollution corresponds to $H_2O$, $CO_2$ and $N_2$ gas that appear at the start of laser heating. This occurs over a few seconds, and we interpret this as the desorption of atmospheric gas adsorbed on the capsule and the surface of the sample holder. The "B" pollution corresponds to a $N_2$ only signal, which starts when the temperature reaches 700-800°C. Experience shows us that this anomaly decreases for blank positions (empty capsule) after several heating episodes, so we

interpret it as a deep degassing of the sample holder. This pollution reappears for each sample and is still there even with a sample holder "cooking" before the start of analysis. However, most of time the pollution signal remains significantly below the $^3$He and $^4$He signal and we thus argue that it does not perturb the analysis to any significant degree. Finally, both pollution effects disappear in seconds to minutes, demonstrating the cleaning efficiency of the traps.

### 4.3.3 Gas consumption correction and $^4$He quantification

We evaluated the correction applied to the $^4$He/$^3$He ratio evolution to take account of the gas consumption (interpolated blank subtraction, sect. 3.2.2, Fig 4B) by looking at both the end trend of samples spectrum (i.e. the static gas measurement) and the global trend of blank spectrum. Indeed, for the first it is expected that, during static gas measurement (Fig. 3 and 4, zone 3), the corrected signal (Fig. 4B and C) will be constant if the correction is well calculated. For the blank test, it is expected that the observed $^4$He/$^3$He evolution will be the same as the predicted one.

About a hundred ramped heating – direct analyses were performed on blanks (empty capsules), standards, and natural crystals between September 2022 and May 2023 (Fig. 12). Considering this dataset using the two criteria described above (trend after correction during the static gas measurement and blank analysis), ~70% of samples successfully pass the two tests and correspond to all samples with significant amount of $^4$He (standard and natural crystals) and ~60% of blanks. However, we observe that for crystals with a low level of $^4$He and ~50% blanks, the linear interpolation does not characterise

the gas evolution appropriately. Practically, for blanks the predicted $^4$He/$^3$He evolution does not fit the observed trend (Fig. 12B) which can be over- and under-estimated. Yet as the observed signal remains mostly linear, we interpret this as an





anormal gas consumption during the first stage (zone 1, Fig. 3 and 4). For low $^4$He crystals, the interpretation is complex as the gas consumption rate competes with the gas extraction rate, but results seem to indicate a potential non-linear gas consumption.

We suggest that crystals which satisfy the two criteria above could be used reliably for $^4$He quantification. For example, 40 Durango (analysed as unknow sample) and 4 MK1 crystals gave successful analyses with the ramped heating protocol and their ages were calculated using the routine calibration and the signal extracted from static gas measurements (Fig. 3 and 4, sect. 3.2.2). The weighted mean ages gave respectively 31.55±0.69 Ma and 17.55±2.6 Ma in accordance with the published values. Given that, individual ages are somewhat dispersed and characterised by high uncertainty (15-20%). This high

uncertainty is directly related to the error in the interpolated blank value that is calculated from the gas consumption (sect. 3.2.2).

Thus, from these preliminary results we argue that the ramped heating – direct analysis approach is satisfactory for crystals with a significant amount of $^4$He. The main limitation in data quality is related to the uncertainty in the gas consumption correction, but we expect that future work should resolve this issue. However, for crystals with a low $^4$He content (below

~$10^{-15}$ mol), the limitation is related to our current experimental setup: the large line volume requires a high voltage filament to analyse residual gas that produce a faster gas consumption making the later faster than the degassing. Therefore, for future development, we recommend a decreasing of the line volume to increase the resolution of low $^4$He content for ramped heating – direct analysis.





**Figure 12: Example of ramped heating - direct analysis results, A: $^4$He/$^3$He signal corrected blank for Durango shard. B: $^4$He/$^3$He signal corrected blank of empty capsules, in grey results possess the expected behaviour after blank correction (flat trend) and in dark anomalous results. C and D, the corresponding temperature evolution of A and B respectively.**

## 5 Conclusions

This contribution reports the development of a new (U-Th)/He analytical protocol made between 2021 and 2023 at the GeOHeLiS plateform, Geosciences Rennes (University of Rennes, France) and we have shared our observations, recommendation and conclusions accumulated over this time. This development work has led to

    (i)     the implementation of $^4$He routine analyses on a quadrupole based on an improvement $^3$He spiking method for the quantification





(ii)      a revalidation of the U-Th-Sm-Ca (and other elements) characterisation approach using standard comparisons rather than isotopic dilution.

These two developments have been evaluated using Durango and MK1 international standard and show a good robustness. Therefore (U-Th)/He ages can be determined with an error less than 7% (1σ) and most of chemical element (major and minor) contents with an error below 1% (1σ). Then, based on our experience we suggest the following:

-      this [4]He quantification methodology proposed in this contribution will readily facilitate modification of purification protocol and/or line structure.

     -      the MK1 crystal is suitable as a secondary standard to evaluate the Durango primary standard calibration.

     -      the chemical analysis protocol developed in this contribution drastically simplifies this part of the method and increases the compositional information obtained on a single crystal (i.e. major and minor elements content).

-      the "U/Th – Sm/Th – Ages" graphical representation may be a useful tool to evaluate the (U-Th)/He protocol and data reliability

Additionally, we have briefly reported the initial results of an in-house "ramped heating – direct analysis" setup and its data reduction. Preliminary experiments show promising results for [4]He rich crystals with the possibility to directly determine from the degassing spectra the degassing fraction (i.e. the diffusion kinetic) and the age of a crystal.

**6    Author contribution:**

AD designed the experiments and built it together with DB, DV and ST. AD, DV, LD and NC participated in the data acquisition. AD developed the Excel® workbook automatization software. GR, MJ and KG were responsible for funding, resources, and supervision. AD prepared the paper with contributions from all co-authors.

**7    Competing interests:**

The authors declare that they have no conflict of interest.

**8    Acknowledgements**

Cécile Gautheron is gratefully thanked for all advice given during the development of this system.

**9    Data availability**

All data final data (Ages, U/Th ratio, weight, REE) can be find in attached excel tables. Ramped heating – direct analyses
results are available upon request regarding the dataset size.



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
