# Peer review of "Technical note: a new analytical protocol for apatite (U-Th)/He and trace element analysis (incorporating a continuous ramped heating measurement system for the He)"

_Geochronology, 2024_

## Referee Comment (RC1)

Review of Derycke et al., (submitted to Geochronology)

This manuscript is a technical paper describing the effort and the development of a (U-Th)/He facility at the University of Rennes (GeOHeLiS laboratory) between 2021 and 2023. This new facility is developing a methodology for Helium (He) released from apatite by continuous ramped heating and a trace element analysis without any spiked solution (i.e. isotopic dilution method), and referred to as the "concentration range approach". Although the ramped heating methodology for He has been previously described and documented, the authors are attempting to use the mass spectrometer sensitivity to determine their $^4$He concentrations based on Durango shards as standards. Overall, I found the manuscript poorly written and organized. The paper has some inconsistencies and a lack of information regarding the trace element analysis as well as the continuous ramped heating protocol. The results and the discussion are overlapping making the paper difficult to follow. Moreover, the authors do not explain the discrepancies between the database and the figures. The new "concentration range approach" methodology is not explained leading to some open questions regarding their protocol. Considering that the "concentration range approach" is presented as a new methodology, I would have appreciated a thoroughly explanation of the authors' work, which is not the case. In addition, the authors' attempt to use the sensitivity parameter from the magnetic sector mass spectrometry to a quadrupole mass spectrometer is not convincing due to a lack of explanation.

The figures are also not quite on the level of publication quality, and the experimental approach and results are not well enough documented. Below I have listed my comments, primarily focusing on the major issues. The formatting aspects and my detailed comments are, at this stage, secondary but need attention. Nevertheless, I appreciate the effort of the authors to try new analytical solutions for noble gas spectrometry and trace element analysis. However, this manuscript is not sufficient to support the authors' methodologies and concepts. Therefore, I do not recommend this manuscript for publication and recommend the authors a major re-work and attention to their manuscript before considering it for publication.

**Major comments:**

**The structure of the paper** is confusing. There is no clear section presenting the methods, the results, or the discussions. As an example, the authors are mixing the description of the noble gas line (section 2 with subsection 2.1) with the $^4$He quantification method (subsection 2.3) and the quadrupole sensitivity determination (subsection 2.4) in a section that should be focused only on the description of their analytical system (section 2). Moreover, the quadrupole sensitivity is further explained in section 4.2 (quadrupole calibration and standard age) and the $^4$He quantification is extended in section 3.2 (U-Th/He protocol) and section 4.3.3 (Gas consumption correction and $^4$He quantification). This structural issue can be applied all along the paper. Consequently, the text is difficult to follow and it is tedious to understand the authors' work. This points toward a lack of attention to the quality of the manuscript. I would recommend the authors re-work their sections and have dedicated sections for their methods and dedicated sections for their results and discussion.

**Use of the Sensitivity for quadrupole $^4$He measurement**. One of the major steps in this paper is the use of sensitivity (S) to determine $^4$He concentration in a quadrupole mass spectrometer. The authors are using the definition of sensitivity, expressed in concentration/signal (mol/amp). The paper is based on the work of Gautheron et al. 2021 where the sensitivity of their VG5400 is determined using such definition of the sensitivity (i.e. in mol/amp). However, the equation provided by the authors differs from the one in Gautheron et al., 2021 as they replaced the Drift (D) parameter with a combination of an ionization efficiency ($I_e$) and a sensitivity term. From my understanding of the Gautheron et al., 2021 paper, the authors used a sensitivity term that differs from the classic sensitivity definition, which corresponds to the capacity of the detector to measure a signal in function of the partial pressure of the gas and is expressed in signal/pressure (usually in amp/torr). Consequently, the sensitivity expressed in the paper (i.e. mol/amp) is not only expressing the mass spectrometer sensitivity (in amp/torr) but also specific parameters to the analytical system (such as the dilution coefficient and the initial extraction volume). Therefore, sensitivity in this paper expresses an extended sensitivity, that is not solely dependent on the mass spectrometer itself, but also on the purification line design. The description given by the authors is, however, resumed at line **97**: "*we built on the Gautheron et al. (2021) method, simplifying it a little and modifying it*" but do not provide sufficient detailed information of how they modify it (see detailed comments below for line **97**), simplify it (I guess by adding two parameters ($I_e$ and S) instead of D), or give detailed information of the concept used. First, I would rather keep the sensitivity as solely a mass spectrometer parameter and not include other specific parameters from the analytical system (the same can apply to the Gautheron et al. 2021 paper). This can be misleading as this formulation of the sensitivity is more complex and more difficult to use for inter-lab comparison and calibration. Secondly, I do not understand why the sensitivity is required here. The $^3$He content from the well-known $^3$He reservoir should be sufficient to calculate the $^4$He content based on the $^4$He/$^3$He ratio measured in the sample (and blank corrected). The authors could cross-calibrate the $^3$He content in the tank using standard Durango shards (similarly to Gautheron et al., 2021) to better account for the non-linearity of the quadrupole signal. Also, their sensitivity measurement remains a bit unclear, as to calculate S the authors simply remove it from equation 2 and consider an ideal behavior of the mass spectrometer (i.e. $I_e$ = 1, implying a constant sensitivity of

the quadrupole over the temperature, the filament voltage, etc.…). If so, equation 2 is then the same as equation 1 given by Gautheron et al., 2021, but without the $^3$He tank calibrated with the Durango shards and the D parameter accounting for the drift. This could lead, to my opinion, to a less precise and less well-calibrated system. This is supported by the range of the Durango He-age determined by the authors (from 23.3 to 38.1 Ma) compared to the ages given by Gautheron et al., 2021 (27 to 34.9 Ma) (see Fig. 8). In addition, the dimension of the equation 2 is quite confusing. If I am correct, the authors are determining the $^4$He content in the sample (mol) by multiplying their sensitivity parameter (mol/amp) to the $^3$He content predicted in the pipette (mol) from their equation 1, and the ratio of the $^4$He/$^3$He measured and blank corrected in the sample. Consequently, the dimension of equation 2 for $^4$He concentration is not in mol, but in mol$^2$/Amp (considering that I$_e$ is dimensionless). At least this is what it is expressed by using the equation 1 and 2 in the paper. What I am supposing, however, is that the authors are using the $^3$He predicted signal (amp) from the pipette solely to derivate the $^4$He signal (amp) from their measured $^4$He/$^3$He ratio. Therefore, knowing precisely the $^3$He content in the pipette seems to not be required (but only the relative decrease over time) as they are using S to calculate their $^4$He content (instead of the $^3$He pipette). I, therefore, do not understand the choice of the authors to not use their well-calibrated $^3$He tank but only rely on Durango shards. In Gautheron et al., 2021, the authors did a cross-calibration of the $^3$He tank content with the deviation of the Durango ages which allows a better constraint on the quadrupole signal variability. That, according to me, would give a much better calibrated system than a calibration solely based on the Durango. In addition, the I$_e$ parameter is calculated (see line 107) by comparing the $^3$He predicted signal in the pipette (mol) to the associated peak height (amp) but according to me, this should be used to convert the signal to concentration and not to determine their drift factor. Moreover, the I$_e$ definition given in the paper seems very close to the D parameter defined in Gautheron et al., 2021, and I do not fully understand the fundamental difference between I$_e$ and D, and how this methodology is simpler or easier to set up, while ensuring similar data quality than in Gautheron et al., 2021. Finally, this section is, for me, not convincing. The authors' explanations are not satisfactory and lack of information. Equation 2 seems wrong in terms of dimension (or incorrectly described) and I am a bit concerned by the data quality that the system can produce due to the uncertainties of the calibration. I am genuinely wondering why the authors are not simply using the protocol from Gautheron et al. 2021 which has a more robust calibration method adapted to the quadrupole line, especially knowing that Cecile Gautheron seems to have helped the authors to set up their system (see the acknowledgment section). If there is a reason behind this strategy is it neither clear nor convincing to me. I would therefore recommend the authors to re-work this section and ensure that all their parameters and concepts are properly explained and tested.

**Calculated age MK1 and Durango.** The validation of the methodology is based on the fact that Durango and MK1 are used as reference standards to check the calibration. As the system is calibrated over a range of 20 (or 36? See detailed comment below for line **122**) Durango shards, it is, indeed, reassuring that the Durango run as unknowns yields to an age in agreement with the calibration using the same Durango age. However, I could argue that this cannot rule out some systematic analytical bias as you are comparing Durango with a calibration based on these very same Durango. On the other hand, the authors provide a dataset on MK1 apatite with a reference weighted mean age given by Wu et al., 2021 Chem. Geo. at 17.99 +/- 0.02Ma (from 191 Apatite fragments and 6 different laboratories). The authors provide a good agreement between this value and their data, with a measured age at 17.7Ma +/- 0.6Ma (over 10 MK1 samples). However, I have few concerns regarding the dataset provided and used in the paper. In the supplementary material (Sup. Mat. 2 – standard ages and ThU, SmTh ratio and Sup. Mat. 3 – standards age calculation) the authors reference 10 MK1 samples providing their age, Th/U and Sm/Th as well as their $^4$He concentration in ccSTP. First, I would like to mention that I would overall recommend avoiding ccSTP as a unit for the concentration. Please use the unit of moles or atoms as it gives a more precise and direct indication of your real concentrations. The ccSTP implies that you normalize the pressure and the temperature of your standard to the normal condition (1 atm and 0ºC) but usually, the temperature and pressure of the standard are rarely given. Therefore, it is a good habit to express the concentration in moles or atoms to be independent of the lab condition and provide to the reader a precise estimation of your measurements. Secondly, the Sup. Mat. 1 – standard concentration, provide 15 MK1 trace element analyses instead of 10 given in the other table and the text. There is no age associated with the extra 5 MK1 samples (which represent 1/3 of the dataset for MK1). I am not sure why those 5 additional MK1 data are disregarded from the dataset. The names of the additional MK1 also overlap with samples already given (there are 2 samples MK1B or MK1C for example) but they display different U, Th, and Sm concentrations, and likely He-age, although no $^4$He concentration or age are provided for these additional MK1 apatite. In addition, the authors specify later that only 4 MK1 were used due to blank limitation (see line 406) but still plotted 10 MK1 ages in Fig. 10. Consequently, the MK1 dataset seems to have some discrepancies that the authors need to clear out, and explain before to use MK1 samples for validation of the methodology. In addition, the errors on the mass, U, Th, and Sm are all constant (for example 2.03% for Ca mass for MK1 and Durango) and I am not understanding why considering the range of content measured in the samples and the fact that the authors are not using any spike (see **Error bar propagation** section below).

**Concentration range approach**. The "concentration range approach" is referred in the text as a well-known method. However, the authors provide no references for the method, only specifying in lines 243-245 "standard concentration ranges to quantify the amount of element in a solution is a classic and well-documented approach in geochemistry". From my experience, I don't know what the term concentration range approach implies. I can only suspect that the authors actually refer to the standard addition method (see *Burns and Walker 2019, Origins of the method of standard additions and of the use of an internal standard in quantitative instrumental chemical analyses.*

*Analytical and Bioanalytical Chemistry* and reference within). If I am correct, I would recommend the authors to be careful with the terminology used, specifically if the authors do not bother to provide references for the methodology. In any case, I would have expected at least a short description of the method used here as it is referred as a new approach to determining trace elements (line 17). I am, therefore, not sure if the authors are referring to the method of standard addition or something else. I would bet, however, that the authors might use the single-point standard addition methods. Consequently, from my understanding, the sample needs to be separated in two, one part analyzed under normal conditions for QQQ-ICP-MS analysis and one part mixed with a known amount of standard (using the dilution factor) and also analyzed with QQQ-ICP-MS. Therefore, if I am correct, how the apatite crystal is prepared/separated for the analysis, what is the spike/solution used in the standard addition method and what is the amount of sample compared to the spiked solution (to avoid matrix effect on the standard's matrix)? It is unfortunate that the authors do not provide much information. The only insights regarding the methodology used by the authors are located in the discussion section and are the minimum concentration required for analysis, and the evaluation of the dilution factor. At this stage, I really would have preferred reviewing what the author did rather than to guess. Despite some apparent encouraging results in Figure 7, there are lots of open questions and uncertainties in this protocol that are concerning. The lack of available data from the authors is also not helpful. For example, figure 7 displays an apparent good fit between the QQQ-ICP-MS analysis and two external measurements (one with LA-ICP-MS and one at the SARM in Nancy) of Durango samples. The authors used this plot as the argument that their "concentration range approach" is valid. Considering the importance of Figure 7, I do not think that the use of a log scale plot is a good visual estimation of how reproducible the method is compared to the other reference techniques. I would therefore question the relevance of CI chondrite for normalization and I would rather suggest a comparison between the three methods, based on the residual spectrum between the external measurement (LA-ICP-MS and SARM) and the new methodology (QQQ-ICP-MS). Such a plot would prove more useful to validate the methods and track down any potential analytical bias or discrepancy. In addition, the authors do not provide the LA-ICP-MS and the SARM data in their dataset or details of the analytical procedure used at the SARM or for the LA-ICP-MS analysis. Finally, a new protocol, even using a well-known methodology requires careful consideration and could justify its own scientific communication. This is, unfortunately, far from what the paper provides. Consequently, the new methodology used by the authors remains unclear. This is very unfortunate as this is critical for understanding how the author determined their U-Th-Sm concentration as well as other trace elements.

**Th/U vs. Sm/Th relationship**. One of the authors' findings in this paper is the linear behaviors of the Th/U vs. Sm/Th in Figure 8. The author suggested that their measured U-Th-Sm ratios are plotted on a negative linear correlation (Fig.8). This negative linear relationship is interpreted by the authors as a product of crystal heterogeneity and they excluded any possible chemical bias on the Th measurement. The authors also report in Figure 8 the age of each Durango shard and show that best age estimations are reached for Apatite crystal lying on the regression line. There is a discrepancy, however, between the available dataset provided by the author (Sup. Mat.1 – standard concentration) and the data represented in Figure 8. To represent that discrepancy, I have quickly plotted (see below) a similar figure to Figure 8 in the paper (i.e. Th/U vs. Sm/Th) but with all the available datasets given in the supplementary material (Figure A) and with highlights on the data only shown in the paper (small red full circles, Figure B).

[Figure]

This discrepancy is not explained in the main text and no reason is given for why not all the provided data is used to interpret the U-Th-Sm dataset. More importantly, the selected data shown in Fig.8 suggest that a clear negative relationship between the Th/U and the Sm/Th ratio exists while considering the whole dataset, this evidence is questionable, especially with the error bars (Figure A). The authors also argue that the (U/Th)-He ages of the Apatite are best when lying on the regression line, however, the apatite ages are not provided for the whole dataset but only for the selected apatite grain (small red full circles, figure B). This led to serious concern regarding the

authors' interpretation of their data. I would suggest using all your available He ages, Th/U, and Sm/Th ratios in one clear figure and verifying if your assumption for a negative linear relationship remains true and if the best ages estimate for the Durango apatite crystals are still located on a specific trend. In addition, I noticed that the range of the Durango apatite ages is quite large, ranging between 23.3 and 38.1 Ma (from Fig.8) while data from Gautheron et al., 2021 are ranging between 27 and 34.9 Ma. This is a significantly larger range, meaning that the quality of the data is also questionable compared to the methods used by Gautheron et al., 2021. This should be mentioned and discussed by the authors. Additionally, the authors ruled out any chemical bias on the Th measurement as the data should then not define a linear regression but a systematic ratio shift. I could argue that a systematic ratio shift from an initial linear trend could still be a linear trend but shifted from its original position. To my opinion, this is not sufficient to rule out any chemical bias on the Th. Moreover, the majority of the U/Th ratios observed from the Durango in Gautheron et al., 2021 display values ranging between 18 and 25, and Sm/Th values ranging between ~0.4 and 0.8 while those analyzed in this paper range between 14 and 20 (U/Th) and ~0.9 and 1.1 (Sm/Th). The significant shift toward higher Sm/Th and toward lower U/Th values is not discussed in the paper but could be the result of a systematic bias on the Th. Systematic bias on Th would induce higher values on the X axis (Sm/Th) and lower values on the Y axis (Th/U) if Th is lost and/or underestimated during analysis. This could explain the shift of the dataset between this paper and Gautheron et al., 2021. Finally, the authors expressed that they already observed such a negative linear trend on volcanic Apatite crystal but failed to provide any references at line 308. I want to point out that "from an internal lab experience" cannot be used as a decent scientific reference.

**Ramped step heating experiment**. The authors claimed that they can derive the diffusion behavior and kinetics information of the samples from their ramped heating experiments (see lines 16, 40, 366, and 444), but this is supported by none of the data provided. The authors did not provide any kinetics information either in the main text or in the figures. I would have expected some $^4$He release spectra in function of the temperature as well as some $^4$He Arrhenius plot of the Durango apatite to investigate if their protocol is reliable (see Idleman et al., 2018). It seems that the authors are relying solely on the comparison between the pyrometer and the thermocouple measurements. However, the methodology used by the authors to mount their capsules with the thermocouple is not properly described (i.e. line 368: "capsule set on a thermocouple"). How the capsule is attached to the thermocouple? do you follow the same protocol given by Idleman et al. 2018? Knowing the critical importance of a well-calibrated temperature system for kinetics studies, it is important to explicitly provide here all the details of the method (see Idleman et al. 2018). Moreover, thermocouples can easily have a bias if not placed properly near the samples (see also Idleman et al. 2018 for detailed discussion). In addition, the pyrometer calibration methodology is not provided by the author (did you calibrate independently the pyrometer against a black body or some reference material?). A better approach, according to me, would have been to compare the kinetics data on their Durango with the literature. The authors, however, only provide a partially calculated diffusion coefficient ($D_0/a^2$) in Sup. Mat. 4, but neither the activation energy (Ea) nor the diffusive behavior of the samples (i.e. plot with $\log(D/a^2)$ vs. 1/T). The $D_0$ coefficient only represents the 'infinite' temperature diffusion and defines the intersection of the $\log(D/a^2)$ axis in an Arrhenius plot. This does not give sufficient insight into the diffusive behavior of $^4$He in their Durango apatite from their ramped step heating experiments. Moreover, the authors provide neither the $^4$He content, the $^4$He signal nor the $^4$He release fraction as raw data in the ramping heating table and therefore make it impossible to investigate any kinetics on their Durango. In addition, the large ramped step heating dataset provided (see Sup. Mat. 4 – ramped heating) is giving zero useful information for the paper despite a large amount of data (see my detailed comments below on Sup. Mat. 4 for more details).

**Error bar propagation**. I noticed that the error bars on the Th/U and Sm/Th are likely underestimated. For example, the authors calculate the error on their Th/U ratio at 4.1% for sample D22P3G while the corresponding error on the U and Th are also given at 4.1% for both U and Th. The authors didn't propagate the error from the Th and the U on the ratio. Usually, an error propagation should lead to an error closer to 5.7% with an error of 4.1% on both the U and the Th. In addition, I would like to know how the errors are calculated for the other trace elements as the errors in the dataset are almost all fixed at a unique value of ~4% for every trace element analysis. I do not understand how you can get a unique and fixed error for all the trace elements analyzed over a wild range of Durango crystals. Please provide more details on your error determination. Similarly, the sample mass (based on the Ca content) also displays a uniform error value (at 2.03% for MK1 apatite as an example), regardless of the amount of Ca measured in the samples. I would have expected some variability at least due to the variation of the Ca mass in each sample and the fact that no spiked solution is used to determine their concentration. I also would like to point out that the authors specified that they analyzed 10 MK1 samples, with a measured age at 17.7Ma +/- 0.6Ma (line 357) but later they mention that only 4 MK1 were successfully analyzed due to blank limitation (line 406) and give an age at 17.6Ma +/- 2.6Ma, which is a much larger error on the analyzed standards. Why there are two different statements on MK1 samples, and which are the MK1 samples that have been rejected over the 10 samples? (or 15 if we refer to the dataset provided in Sup. Mat. 1 – standard concentration, see my comment on the calculated age MK1 and Durango above). This is a bit confusing regarding the importance of the MK1 samples in the validation of the methods.

**Error on the blanks.** In Fig. 4, the authors are extrapolating the cold blank when the system is static, to higher temperature based on the signal drift prior to the experiment. In addition, they also use blank capsules run as samples with the ramped heating protocol (line 395). First how long do you measure the blank evolution prior to starting the laser? And secondly, why do you use a cold blank extrapolation? I would expect that blank capsules

analyzed as samples with the ramped step heating protocol should provide the closest blank values compared to a sample and therefore should be used instead of an extrapolation from the cold static line. Therefore, how do you correct the blank at the end? Are you applying a systematic blank from the extrapolation, or the blank capsules, it is dependent on the sample, or a mix? This remains unclear in the text and the dataset. Please provide more info and a detailed example of how you are treating the blanks. The authors also mentioned the fact that 70% of the samples analyzed on their system (including natural samples, pipettes, and standards) get enough signal to overcome the blank background and "60% of blanks" (line 398). The authors' formulation is unclear. What do 60% of blanks mean here? Is that the sample signal is higher than a blank by 60%? Also, what is the success rate if only natural samples are selected (i.e. excluding your pipettes and standards)? In addition, the authors mentioned some abnormal blank behavior associated with their low $^4$He samples. However, I am not sure why the behavior is qualified as abnormal simply because it is not linear. As this non-linear behavior seems to affect systematically the low $^4$He samples, therefore it should be considered as a normal characteristic of those samples and treated as such. Moreover, it is expected to have a different behavior between low and high $^4$He concentration (or partial pressure of He) in the ionization chamber. Indeed, for low He content, there is a strong memory effect that can contribute significantly to the $^4$He signal, while for high content, this effect is negligible, and mostly gas consumption by the filament affects the He signal. Nevertheless, blank correction should therefore include a systematic non-linear behavior for samples with low $^4$He. In addition, you are smoothing the signal in Fig.4B, which might refer to the filtering process (see also my detailed comment below on Lines 213-214). How that is affecting the initial small $^4$He signal?

**Detailed comments:**

The article uses a mix of French and English words in figures (see Fig.1 with ionique vs ionic) and tables (see Sup. Mat 3 with "Ech" vs. Sample and "masse" vs. mass). This points toward a not thoroughly reviewed manuscript prior to submission by the authors. I recommend the authors again be sure of the quality of the paper prior to submission.

**Sup. Mat. 3** – Standards ages calculation. This supplementary table provides important data from their analyzed Durando and MK1 standards (i.e. $^4$He content (ccSTP), sample mass, $^4$He/$^3$He ratio, pipette number, etc....). The provided data allow the calculation of the $^3$He predicted from each pipette using equation 1 in the paper. By doing such, and considering a Vpip of 5.8cc, a Vtank of 5741cc, an initial $^3$He of $5\times10^{-11}$ mol (given by the author for a cylinder pressure of $1\times10^{-6}$ mbar, at line 103), and the pipette number in the table, we can calculate the $^3$He content predicted for each sample. Alternatively, the $^3$He content for each sample can be determined by comparing the measured $^4$He/$^3$He ratio and the measured $^4$He concentration. However, when I compare the two values, I observe significant discrepancies between the $^3$He predicted by the pipette number and the $^3$He given by the measured $^4$He/$^3$He ratio for the same pipette number. As an example, for the pipette number 180 (sample D22P3G) the $^3$He predicted (equation 1) is calculated at $4.2\times10^{-11}$ mol while the $^3$He estimated from the $^4$He/$^3$He ratio is at $5.4\times10^{-10}$ mol (with a $^4$He/$^3$He ratio at $1.44\times10^{-4}$ and a $^4$He at $7.8\times10^{-14}$ mol). From my understanding, those values should be similar as the $^3$He predicted by equation 1 should be the quantity injected alongside the sample for $^4$He determination. In addition, I observed that the ratio between $^3$He predicted and $^3$He measured is nearly constant over all the standards (i.e. MK1 and Durango), implying that this is likely due to some constant parameter inducing a factor ~13 between the two concentrations. Such discrepancy and homogeneity in that ratio (~13) could reveal an issue with the analytical protocol and/or a miscalibration of the system that requires some investigation.

**Sup. Mat. 4** – ramped step heating data is providing zero useful information. Some of the data name suffixes (i.e. blc, Sci, CO1, or 19#15) are not explained and I do not know what those data represent. Moreover, the sample series D22P2 (from A to T) and D22P3 (from A to F) are not shown in the text or the result or even mentioned or discussed by the authors. The sample series mentioned in the paper (Fig. 10 and Sup. Mat. 3 tables) starts only from D22P3-G (which is exactly when the ramped step heating dataset stops, i.e. D22P3-F). In addition, the raw data only include the $^4$He/$^3$He ratio while it is essential to get the $^4$He signal in function of the temperature (or even the fraction of released He). This is needed if we want to ensure that the ramped step heating protocol produces a normal degassing pattern for diffusion experiments (see my comment above on the ramped step heating experiment). Consequently, I am not sure why the authors provided so much data (up to 27824 lines on the Excel sheet) with zero use for the paper and zero information regarding the samples used in the study.

**Fig. 1**. Please check for the French language occurrence and correct it. What are the blue and red dots on the top view of the sample holder?

**Fig. 9.** How the error bars on the $^4$He calculated content vs. signal are calculated? I suspect it is a plotting effect, but the small $^4$He signals seem to yield to very small error bars (which seems unlikely). I would recommend giving the data in a table in the main text for the reader to understand better the dataset. Also, those data are not provided in the supplementary materials given by the authors. In addition, please explain the label in Figure 9B (what C.H or R.H means?)

**Fig. 10**: The early Durango shards (prior to pipette ~300) are showing much higher heterogeneity in their He-age compared to the one analyzed after pipette number ~300. The author only specifies that this is due to some "Q-He line change" (i.e. analytical system modification). I would like a more detailed explanation of what changed before and after pipette number 300, and how this had such an impact on the dataset. So far this has not been explained.

**Fig 12**. The figure is a bit confusing. Are you using the blank on the empty capsules to correct your signal in all your data or the extrapolation from the static blank evolution? (see my comments above on the error in the blanks). Also, some ramping step heating experiments seem quite erratic with strong variation in the ramping slope. Is that an intentional experiment with extreme change in the ramping or it is reflecting some anomalies during the ramping for some samples? Such anomalies need to be stated and discussed by the author if they are observed.

**Title:** Using parenthesis to refer to the ramped heating measurement is clumsy to my opinion. Everything in a title should be important and concise. Using parenthesis implies secondary information and thus shouldn't be in the title.

**Line 15**: The authors are giving an estimation of their error at "~3.9% in the $^4$He content determination". How this is calculated? The only other occurrence found in the paper mentioning this error is on the Durango ages dispersion over 45 Durango shards (lines 350-351). Regarding the Durango age uncertainties, how do you calculate them as well? From my calculation, the standard deviation calculated from supplementary material (Sup. Mat. 2 – Durango age ThU-SmTh ratio), over the available Durango He-age is 31.1+/- 8.7%. Moreover, this error does not propagate the individual errors from each Durango age which can range from 2.2% (D23P6D) to 16.3% (DG6C). The paper specifies that the Durango age yields to a central age of 31.11+/- 0.7% (or 0.23Ma) which is much less than the standard deviation over the Durango shards analyzed and the minimum error measured on the Durango shard D23P6D. How it is possible? Moreover, as mentioned before, the range of (U/Th)-He age on the Durango Apatite is much greater in this paper than the range observed in the Gautheron et al., 2021 paper. This points toward some analytical issues and the quality of the data is overestimated by the authors.

**Line 66**: Do you refer to a two-color pyrometer or a dual-wavelength pyrometer? Please specify in any case as two wavelengths is confusing with modern pyrometers terminology. Also, how did you calibrate the pyrometer? For dual-wavelengths and/or two-color pyrometers, the emissivity is not required but the slope is an important parameter to calibrate for accurate temperature. Did you use the thermocouple installed on the system and calibrate the pyrometer against it, or did you use a separate system (black body or reference material)?

**Line 88-89**: Not relevant details.

**Line 97**: "modifying a little bit" is neither precise nor adequate to describe what is modified from Gautheron et al., 2021 method for $^4$He determination. I would avoid using such vague descriptions in your scientific papers and rather explain precisely and in detail what you have modified. For the sake of clarity, I recommend that important details should be provided immediately, or within the current section where it is mentioned.

**Line 99**: How do you determine the volume and their associated errors?

**Line 100-101**: replace (re)filled by only filled. Remove the parenthesis statement. This is not relevant as the volume of the pipette is given and it is fixed at ~5.8cc.

**Line 102**: How the internal pressure of the cylinder is adjusted? Does the system have some ballast to modify the internal volume or do the authors purge and fill the $^3$He tank with a more or less diluted standard?

**Line 105**: Equation 1 gives the predicted decrease of the $^3$He content (in mol) in the pipette as a function of the depletion of the reservoir (given by the number of pipettes taken). This equation is correct if the pipette volume is purged every time (i.e. Vpip is pumped out of the left-over gas from the cylinder). Did the authors clean the pipette volume prior to a new inlet? If not, the author needs to consider the left-over gas inside the pipette as it can induce a deviation with time from equation 3 if not accounted for. This effect is important if the pipette volume is large, which seems the case here with a pipette volume of ~5.8cc (vs. ~0.5cc for Gautheron et al.,2021).

**Line 122**: The authors mention ~20 Durango fragments used to set up the calibration, but later in the text they refer to 34 Durango fragments (**line 320**). Which number is the correct one?

**Line 124-126**: The authors specify that they have plotted $^4$He content vs. $^4$He signal for sensitivity determination but no figure is mentioned. I guess the author refers to Figure 9. Please always refer to the figure when you mention it in the text.

**Line 140:** "in-house LabVIEW software that includes automatic grain detection and size measurement" How this is performed automatically with LabVIEW? Does it pick automatically the grain, and record the length, height, and width as well as the termination geometry of the crystal? More details here would be beneficial. I suspect that if using ToupView software, the measurements are done manually. Did you perform a comparison between an automatic measurement with LabVIEW and a user manual measurement?

**Line 170**: You specify that the Q-He line is pumped out for 2 min between samples but at line 198, you mention 10 min. Which protocol is correct?

**Line 210**: Where is the plot for the cumulative helium loss vs. temperature from your data? This should be provided here.

**Line 212**: You mention a very fast transfer time (<1s) between your laser extraction cell and the mass spectrometer. First, the reference provided (McDannell et al., 2018) does not mention or give any indication of the typical conductance time for the $^4$He transfer from the extraction cell to the analyzer. Secondly, the paper from Idleman et al., (2018) does provide some indication and gives a specific time of 2s. They mention: "*In our He extraction line, the time constant for pressure equilibration after the addition of small He aliquots to the extraction line is< 2 s over the range of He pressures typically encountered during CRH experiments. This short delay corresponds to an effective mismatch between the recorded sample temperature and He measurement of 1 °C or less at typical heating rates, and is small enough to be ignored in most situations.*" To my opinion, this needs to be carefully addressed and calculated from your own line. You cannot refer to someone else transfer time (or only compare the range) as this is specifically controlled by the very own (and unique) design of your prep line. Consequently, the time can be a few seconds to tens of seconds and needs to be determined properly. Moreover, you specify at line 415 that your purification line has a "large volume" (although no volume is provided) and therefore this could further increase the transfer time of He due to greater line volume to equilibrate. In any case, less than a second seems an unreasonable short time and I would recommend estimating that time properly and the effect on your He measurement.

**Line 213-214**: How the filter is applied? You mention that $^4$He/$^3$He is the raw ratio, which implies that it is not blank-corrected. Is the filtering made before or after blank correction? This could impact the initial low signal as filtering a dataset implies removing some signal, which can be critical on the first step when the signal is often very close to the noise level. I would recommend preferably to have a measurement of the baseline integrated for each analysis and to subtract the measured noise before any filtering of the data.

**Line 237, 272, 288, 456,** and **Figure 6**: Careful with using weight and mass. I am guessing that you refer to the mass (in g or µg) and not the weight of your sample. Those are two different measures and have scientifically different meanings. Please check your table as well.

**Fig. 7**: Barra et al., 2012 is not listed in the reference. The caption mentions that Durango Apatite "seems to be internally homogeneous". Is that an observation from the figure or an assumption from the authors? Please be more specific and give a value to assert either the assumption or the interpretation.

**Line 237**. The reference for the equation referring to the Gautheron et al., 2021 paper is incorrect. Equation 3 in Gautheron et al., 2021 expresses the $^4$He calculation with their VG5400 magnetic sector field. The correct equation that the authors might refer to is given in Appendix B of the Gautheron et al., 2021 paper (Eq. B5). Gautheron et al., 2021 are using the following equation based on the work of Guenthner et al., (2016):

$$Apatite\ (\mu g) = \frac{^{43}Ca \times 10^{-9}}{0.135/100} \times \frac{1}{0.4 \times 10^{-6}}$$

On the other hand, the authors provide the following more general equation:

$$Apatite_{weight} = \frac{Ca_{weight}}{0.3974}$$

As mentioned before, the authors are likely referring to the mass and not the weight and therefore expressing the mass of Ca and the Apatite in g or µg. Also, the equation does not match the one the article is referring to. Gautheron et al., 2021 are using the $^{43}$Ca isotope to determine the mass, while the authors are using the total Ca content. Could you please provide the correct references, and detail why you are using total Ca and not the isotope $^{43}$Ca to determine the Apatite mass? How does this impact your mass determination and the effect on the U-Th-Sm?

**Line 350**: you provide an average value of 31.1Ma with an error of 0.23Ma based on the 45 Durango fragments analyzed. In the supplementary material (Sup. Mat. 2 – Durango age) you provide 60 Durango samples but you do not specify which ones are selected for the average age. Nevertheless, I am guessing that you removed the Durango shards analyzed prior to pipette #319 as before that, the Q-line seems to have analytical issues (those issues are, however, not described or specified in the text). Therefore, taking the data of Durango shards from pipette #335 to #874 led to an average age of 31Ma +/- 1.6Ma (or 5.2%) for the standard variation. How do you calculate your age with an error of 0.23Ma (or 0.7%)?

**Line 351**: What means 3.9+/-1.3% (is 3.9Ma +/-1.3%)?

**Section 4.3**. The title of the section is confusing (what is feedback on a ramped heating development?). Therefore, the subsections are lacking of homogeneity (see my comment on the structure of the paper above).

**Line 386**: "Sample holder cooking"? Do you mean baking?

**Line 402**: How the first stage of the laser run (zone 1) is inducing abnormal gas consumption? He signals from Fig. 3 seem to not be affected by the burst of gas observed in the first stage. Also, the timing of the gas burst in zone 1 in Fig. 3 seems to be on the order of ~1 min while non-linear behavior for He in Fig. 12 seems to last up to ~10min (the time on the X axis in Fig. 3 and 12 are hard to read however). Consequently, that assumption from the authors seems unlikely.

This detailed list is not exhaustive and can include more comments. However, I think the authors get some ideas of the work required for the paper. I am, therefore, stopping here the detailed comments but strongly advise the authors to thoroughly check and ensure a sufficient quality of every section and data presented in the paper before publication.

---

## Author Comment (AC3)

**Global response to the reviewers – Gchron 2024-6**

The manuscript Gchron 2024-6 was reviewed by four different reviewers (R2: James Metcalf, R4: Bruce Idleman and R1-3 anonymous), who are greatly thanked for the time they took to do so and for the extensive comments and recommendations they provided. These reviews contain similar or related comments, and in the following we will attempt to address and clarify those general comments.

*To supplement this overarching response*, **if the manuscript is accepted for review,** *individual and precise replies will be produced as responses to each review on the Gchron 2024-6.*

Our general response will initially address the scientific remarks and issues before moving on to the manuscript structure comments. We conclude with a proposal for a revised version of the manuscript.

**SCIENTIFICS COMMENTS:**

**NOBLE GAS QUADRUPOLE MASS SPECTROMETER CALIBRATION AND USE:**

R1, R2, and R4 each provide significant comments, criticisms, and inquiries regarding the approach outlined in the manuscript for calibrating and measuring the $^4$He gas. We will address the scientific aspects of these comments. However, they also raise the question "why develop this approach when a better one already exists?" This legitimately challenges the relevance of the proposed manuscript and will therefore be addressed separately in the last part of the reply ("Future of the manuscript").

From our perspective, one of the key comments about the noble gas protocol concerns equation 2, particularly the parameters S and $I_e$. Reviewers R1 and R4 highlighted numerous aspects of this equation that may lead its misinterpretation and, consequently, its calibration and use for noble gas content. These comments rightly challenge our presentation of the protocol, and we recognize that the original manuscript is lacking detail and clarity in several places. The following paragraphs will attempt to rectify these problems.

In a revised manuscript, the following information would be added to the Introduction section (to clarify the context and purpose of the article) and to the methodological section.

➢ Technical justification for development of the protocol

The original manuscript failed to mention a practical constraint that led us to develop our protocol. During development, the laboratory did not have sufficiently accurate pressure sensors with a wide enough pressure range, so we were unable to accurately assess the different volumes in the line (up to ~20% error for some volume determinations). This then limits accurate determination of the partial pressure or amount of $^3$He that will reach the quadrupole mass spectrometer using a classical approach based on the line volume derived from parameters such as pressure, volume, and number associated with the $^3$He tank and pipette.

However, the experience gained by A.D. during his work at the GEOPS laboratory between 2017 and 2021, lab operating for over a decade (see Gautheron et al., 2021), suggested that it would be possible to address these limitations.

*At this juncture, A.D. must acknowledge that the development presented in the article cannot solely rely on the research conducted between 2021 and 2023 at Geosciences Rennes, but also draws from the experience acquired beforehand.*

> ➤ Missing formal hypothesis:

Consequently, we suggested that for the same "analytical set-up" (i.e., sequence of volume opening/closing/cleaning), the amount or partial pressure of the $^3$He spike reaching the quadrupole mass spectrometer for analysis will be constant (except for the spike, which decreases with use). Therefore, by analyzing the $^3$He signal (in amperes) together with the signal (also in amperes) of different **known** amounts of another gas (in partial pressure or moles), it is possible to calculate the amount of $^3$He spike (in partial pressure or moles) in the quadrupole. The calculated value could then be used instead of one derived using line volume and retained if the "analytical set-up" remains unchanged.

Given this approach, we suggest that the obtained $^3$He spike content will be determined with the same precision as the known amount of the other gas used. Below we address the relevant comment from R4

R4: "*I question the validity of using a material with a somewhat poorly constrained age like Durango apatite as a primary standard for helium measurement calibration – it can be done, but it introduces both additional complexity and uncertainty.*"

While we agree with R4's comment, this will be further discussed in the last section ("Future of the manuscript"). However, the explanation below provides arguments to support this discussion.

> ➤ Applying the hypothesis of external volume calibration:

For spike calibration, various known amounts of gas are typically required, and the conventional approach involves using a precisely calibrated tank/pipette with a precisely known amount of $^4$He gas and varying its $^4$He amount thank to the different line characterized volumes. However, due to the difficulties in estimating our line volume we propose using different-sized fragments of Durango as different "$^4$He tank sizes".

We acknowledge that the Durango crystal is far from perfect, it is one of the few internationally recognized standards for $^4$He content (McDowell et al., 2005, Wu et al., 2021). Given this, we decided to assess its use as a calibration standard for our protocol.

The usual equation used to determine $^4$He content is:

$$^4He\ [mol] = \frac{^4He\ _{blanc\ corrected}\ [signal\ or\ partial\ pressure]}{^3He_{blanc\ corrected}\ [signal\ or\ partial\ pressure]} \times {}^3He_{pip}\ [mol] \qquad (1)$$

where the $^3He\ _{pip}$ is determined using the classical approach (tank pressure and volume, pipette volume and number and line volumes).

We invert the equation to give:

$$^3He_{pip}\ [mol] = \frac{^4He\ _{blanc\ corrected}\ [signal\ or\ partial\ pressure]}{^3He_{blanc\ corrected}\ [signal\ or\ partial\ pressure]} \times\ ^4He\ [mol] \qquad (2)$$

In this case, the $^4$He directly comes from a given Durango crystal and is subsequently determined using that crystal's uranium (U), thorium (Th), and samarium (Sm) content, as well as recognized ages (31.02±1.01 Ma, McDowell et al., 2005). This method enables us to calculate the $^3$He$_{pip}$ content for a specific pipette and in principle allows us to implement Equation 1. However, four significant questions remain...

**Question 1: Does this calibration remain valid over time?**

If we repeat this operation multiple times the $^3$He $_{pip}$ spike pressure will decrease following a known equation ($\frac{tank\ volume}{tank\ volume+pipette\ volume}^{pipette\ nb}$) and the calculated $^3$He$_{pip}$ should similarly decrease. In our dataset we get, as is seen in our data:

[Figure]

This observation confirms the approach's expected behaviour over time but **does not** validate the $^3$He$_{pip}$ calculated amount in mol.

**Question 2: Does this calibration remain valid for various $^4$He contents and therefore variable partial pressures?**

We investigated if varying the amount of $^4$He gas affects the calibration result (calculated $^3$He$_{pip}$ ). To do this, it is necessary to obtain correct or eliminate the impact of the spike decrease. Therefore, we propose modifying Equation 1 by replacing $^3$He pipette [mol] with $^3$He pipette [signal], corresponding to the signal obtained for a pure $^3$He spike measurement. Consequently, we obtain a similar form as Equation 1, but $^4$He [mol] is substituted with a corrected $^4$He [theoretical signal] adjusted for partial pressure variation and the decrease in $^3$He$_{pip}$.:

$$^4He\ [theoretical\ signal] = \frac{^4He\ _{blanc\ corrected}\ [signal]}{^3He_{blanc\ corrected}\ [signal\ ]} \times\ ^3He_{pip}\ [signal] \qquad (3)$$

where the $^3He_{pip}$ [signal] is given by:

$$^3He_{pip}\ [signal] = \frac{tank\ volume}{tank\ volume + pipette\ volume}^{pipette\ nb} \times\ ^3He_{initial}\ [signal]$$

where the $^3He_{initial}$ [signal] corresponds to the first signal obtained for a pure spike analysis and the $\frac{tank\ volume}{tank\ volume + pipette\ volume}^{pipette\ nb}$ can be determined by:

- using the respective volumes (if known)
- monitoring the evolution of $^3He_{pip}$ [signal] by doing multiple pure pipettes analyses and determine the value by regression.

By implementing this modification, it becomes feasible to calculate $^4$He [theoretical signal] and compare it with the actual $^4$He [mol or ccSTP] derived from different-sized "Durango tanks". If the variation in $^4$He content from different sources does not affect the calibration, the resulting relationship should be linear with the y-intercept should be 0.

[Figure]

This obtained parameters confirm that a variation of $^4$He content (partial pressure) does not significantly affect our calibration and allows us to convert the $^4He$ [theoretical signal] in a $^4$He amount [mol]. In the case the intercept =0, the slope corresponds to the parameter "$S$" in the original manuscript.

We propose reorganizing eq. 1 as below (eq. 2 in the original paper):

$$^4He\ [theoretical\ signal] = \frac{^4He_{blanc\ corrected}\ [signal]}{^3He_{blanc\ corrected}\ [signal\ ]} \times\ ^3He_{initial}[signal] \times \left(\frac{Vol_{cylinder}}{Vol_{cylinder} + Vol_{pipette}}\right)^{pipette\ nb} \quad (4)$$

$$^4He\ [mol] =\ ^4He\ [theoretical\ signal]\ \times S$$

In essence, this formulation is not fundamentally different from Equation 1, but we believe that it explicitly identifies the parameters ($^3He_{initial}[signal]$ and S) that require modification or redetermination in case of changes in the "analytical step-up" (such as adjustments in volume or spectrometer tunes).

R1 and R4 noted that the choice of "S" as a constant name for the parameters calculating the $^4$He [mol] from the $^4$He [theoretical signal] is not ideal, as it may cause confusion with sensitivity parameters used for magnetic sector spectrometers. We agree with this observation, and for any future manuscript submissions, we will consider changing its name to "T" for theoretical calibration.

Finally, we emphasise that by selecting crystal fragments of different sizes, we cover a wide range of total $^4$He content and therefore a wide range of partial pressures. This makes it possible to validate a wide range of calibrations using only "Durango tanks".

**Question 3: Is the inferred $^3$He content accurate?**

Given eq. 4, this parameter is not used (or at least not directly) to quantify the $^4$He content of a sample. It is replaced by other parameters: "S", $^3$He$_{initial}$ and the evolution of calculated $^3He_{pip}$ (pipette function) and for all these parameters it is possible to define their uncertainty:

-   *"S"*: calculated using the regression results from $^4$He [theoretical] error (gas analytical uncertainty) and $^4$He content error (Durango ages uncertainty + chemical analytical uncertainty)
-   $^3He_{initial}[signal]$ : correspond to the analytical uncertainty of the first pure $^3$He spike analysis
-   Pipette function ($\left(\frac{Vol_{cylinder}}{Vol_{cylinder} + Vol_{pipette}}\right)^{pipette\ nb}$ ): in our case this is calculated from the tank and pipette volumes (only one that could have been determine externally using a a membrane pressure gauge, MKS902, an know volume). Note that, this function, and associated error, can also be derived from $^3$He spike signal evolution.

Following the comments of R1 and R4, the sum of these errors is the one that should be compared to the $^3He_{pip}$ [mol] in case of a conventional analysis. In a new submission of the manuscript, this will appear in the discussion.

**Question 4: Is the obtained $^4$He content for a conventional sample reliable?**

To address this question, we want to analyze crystals of other standards. Assuminf that the U, Th, and Sm content are well characterised, any discrepancies in the characterization of $^4$He would be manifested as a deviation from the expected age. Due to the absence of international standards, we only conducted this analysis on the MK1 standard (Wu et al. 2021), which yielded successful results. However, we argue that since each fragment of Durango varies in total $^4$He content, each one could be seen as a "new blind" $^4$He standard each time.

> ➢ Routine use of the method:

Using this approach on a day-to-day basis with a consistent analytical setup revealed that the calibration performance (represented by the "S" parameter) may become inaccurate after unexpected events (such as power failures or changes in atmospheric conditions) or after prolonged periods of analysis or inactivity (several months). This phenomenon is rarely addressed

in publications, one notable exception is Gautheron et al. 2021, which proposes incorporating this phenomenon using a parameter "D" (for "Drift") in Equation 1 and manually calibrating it using regular Durango analyses. For the approach presented in our manuscript, there are two ways to address this calibration inaccuracy (drift):

1. Recalibrate the "S" parameter regularly (e.g. every month)

2. Incorporate the "D" parameter from Gautheron et al. (2021) into Equation 4 and only recalculate the "S" parameter in case of changes in the analytical setup (such as a change in purification process).

Equation 4 modification: $\ldots \times \left( \frac{Vol_{cylinder}}{Vol_{cylinder} + Vol_{pipette}} \times D \right)^{pipette\ nb}$

After comparing the two approaches for the same dataset over ~2-3 months, it appears to us that recalibrating S is preferable: the "S" parameters changed one time and it was stable, whereas "D" was modified 2-3 time and seem to be unstable. However, this is just a feeling and to obtain statistically indicative data it would be necessary to carry out this work over a longer period.

This feeling is consistent with how the "D" parameter functions within the equation by playing inside the pipette decrease. Indeed, the "D" construction seems incompatible with the evolution of the signal (in amps) of the $^3$He pipette analysis over time, which demonstrates good predictability using the equation.

To clarify this, the four following charts represent the $^3$He signal evolution again the pipette number using the same approach as for Fig. 9B in the original manuscript. It presents the real $^3$He signal (in blue), the $^3$He predicted evolution using the equation including D (in green) and using the equation without D but a "S" modification (in red and simulates a Ie parameter change). Additionally, the last chart (in grey) display the same approach but with the $^3$He signal related to the data and line from Gautheron et al. 2021. It appears that playing on the D parameter (green curve) change significantly the predicted $^3$He evolution, but not in a "linear" way. This aspect of the D parameter implies a frequent recalibration in case of any small variation or derive from the initial calibration.

[Figure]

However, the simplicity of using the "D" parameter during daily analysis is appealing. Therefore, we adapted it to our approach (case 1) by incorporating a multiplicative factor to the "S" parameter. This is the "Ie" parameter in our orginal equation 2. It can be calibrated using Durango similarly to the calibration process for the "D" parameter. From our perspective, this offers a simple way to incorporate the "Drift" phenomenon but linking it more closely to variations in intra-quadrupole mass spectrometer parameters (such as partial pressure, voltage stability, and filament life), than to variations in $^3$He tank and pipette volumes.

[Figure]

To conclude the "**Noble gas quadrupole mass spectrometer calibration and use**" authors would like to highlight that at the end, the obtained equations are not so different from the classical one. Irrespective, our method circumvents the need for line volume calibration by relying solely on the quadrupole mass spectrometer signal and Durango fragments. Therefore, it may be useful to researchers who have similar technical limitations to ours. However, demonstrating how to successfully apply the approach requires more discussion which we return to later ("Future of this manuscript").

**RAMPED HEATING PROTOCOL:**

Reviewer R1 and R4 provided detailed comments about the ramped heating methodology, and the following paragraph aims to address them. They expressed concern about the quality of our temperature monitoring calibration. We use a two-wavelength industrial pyrometer (Endurance® Series – E2RL-V0-0-0) sold by Fluke Process Instrument™ to monitor the temperature of the packet during heating. This pyrometer is certified to measure the temperature of metallic surfaces with a precision of ±2°C. While this industrial certification provides an initial guarantee, the details of the calibration process are proprietary, and therefore inaccessible to the authors. Consequently, we conducted our own test, as presented in our original manuscript.

The test involved using a thermocouple K type - class 2 with an extremity diameter of 1mm, threaded inside the packet of 1mm diameter (and 1mm height). Given the packet and fibre diameter, we estimate that the temperature coupling between the packet and the pyrometer was nearly perfect.

[Figure]

Consequently, we argue that the results presented in the original manuscript are reliable within the discussed limits in the original manuscript (±6°C above 300°C), that is far above what's needed for conventional analysis but not for ramped heating. In response to R4's recommendation, we will replace Fig. 11.B with a chart presenting the temperature difference vs temperature. R1 proposed adding a third calibration by conducting a CRH analysis on a Durango crystal and then using published diffusion parameters of Durango (D0 and Ea) to reassess the measured temperatures. While an interesting idea, we believe this solution may be inefficient, as the Durango diffusion parameters are not easily reproducible, as noted by R4, and it would introduce additional uncertainty from the mass-spectrometer analysis and the kinetic parameter uncertainties.

R4 also highlighted the limitation of our current pyrometer that does not allow precise temperature determination below 350°C. This is a crucial range for kinetic parameter determination, especially for apatite where temperatures between 25-250°C are most significant to study crystal damaging/annealing effects. As in the original manuscript, we acknowledge this limitation as a significant issue, which is one of the main reasons why CRH analyses are not more developed in this publication, and **why we did not present kinetic parameters**. However, we believe that the technology presented in the manuscript holds promise, particularly with the introduction of the new E3ML pyrometer (Fluke Process Instrument™) with a range of 50-1000°C, though we have not had the opportunity to test it. To our knowledge, these methodological perspectives have not been published to date, therefore, the results presented in the manuscript offer limited information but provide a first insight into the reliability of this technology.

In case of re-submission, a paragraph will be added to the discussion to clarify and discuss this aspect.

**"CHEMICAL PROTOCOL" PART (SECTION 3.3 IN THE ORIGINAL MANUSCRIPT):**

This section in the initial manuscript was unclear for Reviewers 1 through 4, with Reviewer 3 specifically requesting additional clarifications. Therefore, for a revised manuscript, we will clarify and add details on this part of the protocol.

However, we believe that it is not necessary to present the chemical protocol in great detail, as it is already published (for instance Tharaud et al. and reference therein) and routinely performed in many laboratories (for instance the SARM at CRPG, Carignan et al. 2001). Nevertheless, we will add some details as in the following paragraph (initial text in black and the added text in red).

*"After the degassing, capsules are retrieved and put in individual 10mL vials (INFO) for chemical digestion and elemental quantification. This quantification is done by signal comparison between dissolved apatite solutions and a range of standard solutions with adapted concentration. In addition to U, Th and Sm concentrations used for helium age calculation, all REE elements are routinely quantified to provide complementary information for interpretation on each grain (inclusion, sample homogeneity, source for detrital samples). Ca is also systematically analyzed to determine apatite weight, considering Ca is stochiometric and a fluorapatite composition (Eq. 3, Gautheron et al., 2021).*

$$Apatite_{weight} = \frac{Ca_{weight}}{0.3974} \qquad (3)$$

*The range of standard solutions is prepared using mothers' solutions (Inorganic Ventures®, CMS1 and CGCA) characterized by precisely known element concentrations (uranium (U), thorium (Th), samarium (Sm), rare earth elements (REEs), and calcium (Ca)) with ISO 17034 and ISO 17025. Those mothers' solutions are combine and diluted using a 0.5N HNO$_3$ solution (distilled from HNO$_3$ 65N – Normapur® VWR®) to obtain 10 to 13 daughter solutions with varying concentration to cover the expected range in dissolved apatite. Specifically, we target concentrations between 0.1 ng.l$^{-1}$ (or ppt) and 10,000 µg. l$^{-1}$ (or ppb) based on a calculation of dissolving a spherical apatite grain with a radius of 50µm (typical size of analyzed apatite) in 2 to 10 mL (required analytical volume), considering U, Th, Sm, REE concentrations ranging between 1 to 500 ppm and a stoichiometric Ca input (~1%).*

*The apatite digestion protocol is adapted from Farley (2000) and Gautheron et al. (2021) and is performed by 100µl acid attack (HNO$_3$ at 5N or 27%, – distilled from HNO$_3$ 65N – Normapur® VWR®) on a hot plate set to 65°C during 3 hours. After 30 minutes of cooling, solutions are then completed for QQQ-ICP-MS analysis with a HNO$_3$ – 0.5N or 2% solution (distilled from HNO$_3$ 65N – Normapur® VWR®) to 2 to 10 ml depending on the required concentration and analytical protocol. The micropipette used for the acid attack and dilution come from Eppendorf Research Plus® respectively 100-1000µl and 0.5-5 ml. After the analysis the platinum capsules are retrieved and returned to the supplier for recycling.*

*The elemental characterization in solution is made on an Agilent 8900 QQQ-ICP-MS. During analysis, the full standard solutions are analyzed alongside sample apatite solutions at regular intervals, typically all 10 for every five apatite solutions. The obtained raw signals are then used to determine the concentrations of apatite solutions using regression (e.g. Tharaud et al., 2015). To calculate the element content for each individual apatite, obtained apatite solutions*

*concentrations are then multiply by the dilution factor. The dilution factor of each digestion session is determined using the micropipette volumes and uncertainties. For each chemical session, the micropipettes volumes are checked by randomly weighing 10 apatite solutions after dilution. The reliability of our element quantification has been assessed by analyzing fragments of homogeneous apatite crystals previously characterized by the same method in a certified lab (SARM, ISO 5 and 6), and by another method (LA-ICP-MS) at Géosciences Rennes (see Section X.X for the results). Results of this test will be discussed in Section 4.4.*

*To assess the stability/reproducibility of these protocols, we analyze Durango and MK1 crystals as if they were unknowns, and then use it to control the obtain results. This choice was logic as they are already routinely measured for (U-Th)/He age calculations."*

**FORM COMMENTS:**

**MANUSCRIPT ORGANIZATION:**

All reviewers emphasize that the manuscript is poorly organized and lacking details, leading to difficulties in following and understanding it:

Reviewer 1: *"The manuscript is poorly written and organized. The structure of the paper is confusing, lacking clear sections for methods, results, or discussions. The text is difficult to follow."*

Reviewer 2: *"Areas where the contribution could be greatly improved with additional clarifications, discussions, and modifications. A re-organization would be beneficial."*

Reviewer 3: *"I cannot support the publication in its current form. My main concerns relate to the organization of the manuscript and missing details in some descriptions. Sections 2-3 could be combined under a Methods section, and separating Results and Discussion would be helpful."*

Reviewer 4: *"A lack of organization makes the discussions difficult to follow. I suggest considering a major reorganization of the text. Additionally, the description of instrumentation and basic analytical methods should not be mixed with that addressing data treatment and analytical complications."*

To address this, we propose restructuring the manuscript, primarily based on the suggestions from Reviewer 3:

- Expand the Introduction (Section 1) to briefly outline the current "classical" analytical protocol and the associated aspects/challenges that will be addressed in the paper leading to our proposed approach. This addition will also facilitate a discussion about the benefits and drawbacks of our protocol, as suggested by Reviewer 4. Additionally, we will contextualize the development and objectives of this protocol to include details provided in the Scientific reply
- Merge Sections 2 and 3 into a single "Methods" section, comprising two main subsections: a "Noble Gas Protocol" and a "Chemical Protocol," as recommended by Reviewer 4:

*"I suggest that the authors consider a major reorganization of the text, at a minimum separating the sections dealing with the He mass spectrometry and those describing the LA-ICP-MS measurements for U, Th, and Sm."*

The new "Chemical Protocol" sub-section will be expanded based on the details provided in the Scientific reply.

- Divide Section 4 into two separate sections, "Results" and "Discussion," structured similarly to the Method section, in accordance with the advice from Reviewer 3: "

"*Furthermore, it would be helpful to have the same (similar) subsection headings in Methods and Results so that one knows where to find the results (like you have it for Sections 3.2.2, 3.2.3 and 4.3 concerning ramped heating).*"

• Introduce a subsection in the "Discussion" where the advantages and disadvantages of the presented protocol will be evaluated in comparison to existing protocols, as recommended by Reviewer 4.

Given these recommendations, we will restructure these parts of the manuscript accordingly

**Language and style:**

All reviewers have pointed out the presence of French words in figures and tables, which will be thoroughly reviewed and corrected for any subsequent manuscript submission.

**Inconsistency between the supplementary material data, figures and main text:**

The main author (A.D.) extends sincere apologies for the inconsistency identified, which solely resulted from an oversight on his part. The supplementary data underwent re-editing after the initial submission, as detailed in the discussions on GChron, to incorporate additional data originally intended for "upon request" availability. During this process, the main author transitioned to another laboratory, and the overly rapid retrieval of the raw data compounded by human error, led to mixing of conventional, ramped heating, and development/test data.

For any revised manuscript, we commit to thoroughly reviewing and ensuring consistency between the manuscript and the supplementary material.

R1: several mention to this aspect in it "Th/U vs. Sm/Th relationship:" section

R3: several mention to this aspect in it "Inconsistency of results and data presentation:" section

**FUTURE OF THE MANUSCRIPT**

The different reviews highlighted scientific errors and structural ambiguities in the original manuscript. We acknowledge and largely agree with these criticisms and have aimed to address them in this response. However, the general question remains regarding the purpose and scientific significance of the article, essentially asking, "Why develop this approach $^4$He calibration when a good one already exists?" To address this question, we wish to provide context for our work.

The (U-Th)/He method has become a routine technique (for approximately two decades) but there is ongoing developments to enhance its resolution and introduce new perspectives. Despite this extended period of use and the presence of well-established labs, there seems to be a lack of openly available publications describing a complete analytical protocol. This observation, not intended as criticism, suggests perhaps a potential blockage when establishing new laboratories and the prospect of losing acquired knowledge and cumulative feedback from different labs. While of course there are forums for exchange (e.g. the Noble Gas network) we view the GChron – Technical Notes as a more formal route to make analytical protocols publicly available. We note that reviewer 2 made a similar observation:

*"As a (U-Th)/He lab manager I found this paper very interesting, and am very supportive of it, and other technical descriptions of analytical set ups, appearing as technical notes in EGU Geochronology."*

In this context, the study presents the results of a first (U-Th)/He analytical development in a new laboratory at Rennes University. However, after two years of development, A.D. who developed the protocol moved on and the laboratory underwent renovations that halted its analytical capabilities. The two-year period was sufficient to test and establish an initial version of the protocol and conduct some routine analyses, but not enough to refine. We believe that valuable information and knowledge was accumulated during this two-year development process, such as the possibility to perform the U, Th, Sm and Ca in a simpler way than isotopic dilution, the potential of using multiwavelength pyrometer for temperature determination rather than a thermocouple and the possibility to avoid volume calibration.

The authors acknowledged that all this development (especially regarding volume calibration) does not allow for obtaining the same "quality" as the conventional approach and therefore led to a "step back" in precision. However, from a practical perspective, the lab has been running for less than a year, and this short period of time leads the authors to assume that with more time, it may be possible to significantly improve precision and reaching the conventional "quality". To our knowledge, these methodological perspectives have not been published to date, and as the analytical laboratory was turn down for an undetermined time, authors believed that it would contribute to the community to share those developments and there is the reason of this article.

Finally, we did not write the manuscript to prescribe our specific approach in preference over others but rather to present its development and our initial results. Over time, as is normal, the scientific community will determine its usefulness. We recognize that a published article must adhere to ethical and scientific standards, including a reproducible protocol, and that the peer review process is in place to evaluate this. Therefore, they will understand if reviewers and editors conclude that the study does not meet these standards and decide to reject the original

manuscript. In any case, authors express gratitude to the reviewers for their valuable feedback and help in organizing their thoughts, hoping that the dialogue remains open.